# Dual-functional peptide with defective interfering genes effectively protects mice against avian and seasonal influenza

Hanjun Zhao[1,2], Kelvin K. W. To[1,2,3], Hin Chu[1,2], Qiulu Ding[4], Xiaoyu Zhao[2], Cun Li[2], Huiping Shuai[2], Shuofeng Yuan [1,2], Jie Zhou[1,2], Kin-Hang Kok [1,2], Shibo Jiang[5,6] & Kwok-Yung Yuen[1,2,3,7]

Limited efficacy of current antivirals and antiviral-resistant mutations impairs anti-influenza treatment. Here, we evaluate the in vitro and in vivo antiviral effect of three defective interfering genes (DIG-3) of influenza virus. Viral replication is significantly reduced in cell lines transfected with DIG-3. Mice treated with DIG-3 encoded by jetPEI-vector, as prophylaxis and therapeutics against A(H7N7) virus, respectively, have significantly better survivals (80% and 50%) than control mice (0%). We further develop a dual-functional peptide TAT-P1, which delivers DIG-3 with high efficiency and concomitantly exerts antiviral activity by preventing endosomal acidification. TAT-P1/DIG-3 is more effective than jetPEI/DIG-3 in treating A(H7N7) or A(H1N1)pdm09-infected mice and shows potent prophylactic protection on A(H7N7) or A(H1N1)pdm09-infected mice. The addition of P1 peptide, which prevents endosomal acidification, can enhance the protection of TAT-P1/DIG-3 on A(H1N1)pdm09-infected mice. Dual-functional TAT-P1 with DIG-3 can effectively protect or treat mice infected by avian and seasonal influenza virus.

[1] State Key Laboratory of Emerging Infectious Diseases, Li Ka Shing Faculty of Medicine, The University of Hong Kong, Pokfulam, Hong Kong. [2] Department of Microbiology, Li Ka Shing Faculty of Medicine, The University of Hong Kong, Pokfulam, Hong Kong. [3] Carol Yu Centre for Infection, Li Ka Shing Faculty of Medicine, The University of Hong Kong, Pokfulam, Hong Kong. [4] Department of Pathology, Li Ka Shing Faculty of Medicine, The University of Hong Kong, Pokfulam, Hong Kong. [5] Key Laboratory of Medical Molecular Virology of Ministries of Education and Health, Shanghai Medical College and Institute of Medical Microbiology, Fudan University, Shanghai 200032, China. [6] Lindsley F. Kimball Research Institute, New York Blood Center, New York, NY 10065, USA. [7] The Collaborative Innovation Center for Diagnosis and Treatment of Infectious Diseases, Li Ka Shing Faculty of Medicine, The University of Hong Kong, Pokfulam, Hong Kong. These authors contributed equally: Hanjun Zhao, Kelvin K. W. To. Correspondence and requests for materials should be addressed to K.-Y.Y. (email: kyyuen@hku.hk)

Seasonal influenza virus annually causes over 3–5 million cases of severe illness with about 0.25 million deaths globally. Antigenically-shifted zoonotic influenza viruses pose threats of another pandemic[1–3]. Neuraminidase inhibitors, such as oseltamivir and zanamivir, have been used clinically for many years. However, human isolates of A(H1N1)pdm09, A(H3N2), A(H5N1), and A(H7N9) resistant to neuraminidase inhibitors have been found[4–7]. Convalescent blood products with high titer of specific neutralizing antibody have been shown to improve survival, but are not readily available[8]. Thus, broad spectrum antivirals with low possibility to induce resistance are urgently needed for controlling influenza virus infections.

Defective interfering (DI) viruses, which are generated naturally during viral replication with internal deletions in viral genes[9,10], can compete with the growth of wild-type virus and therefore suppress the replication of wild-type virus by interfering with the expression of the cognate full-length RNAs[9,11,12]. Though influenza DI virus (DIV) has been shown to be effective in vivo as a potential broad-spectrum antiviral with low risk for inducing resistance[13–16], there are several concerns of influenza DIV used as therapeutic agents. Firstly, influenza DIV may reassort with wild-type influenza A virus to generate novel reassortants[17]. Secondly, neutralizing antibody may develop against the DIV and affect the antiviral effect in subsequent use. Delivering defective interfering genes (DIG) as an antiviral may avoid the risk of generating new reassortant virus and the problem of unwanted immunogenicity.

In this study, we investigated the use of three DIG (DIG-3) as an antiviral in the treatment of influenza virus infection. In the first part, we confirmed that DIG-3 of influenza A virus PB2, PB1, and PA genes could efficiently inhibit influenza A virus replication in vitro. Transfection of DIG-3 in vivo by jetPEI could significantly protect mice from lethal A(H7N7) virus challenge. In the second part, we further improved the in vivo antiviral efficacy of DIG-3 by using a dual-functional peptide vector. This dual-functional peptide vector consists of two components, HIV-1 Tat (TAT) and P1 peptide. TAT is a peptide widely used for in vitro and in vivo transfection[18–21]. P1 peptide is a derivative of an antiviral peptide P9, which we have previously designed based on the mouse β-defensin 4 and was identified to have antiviral activity against influenza A virus H1N1, H3N2, H5N1, and H7N7[22]. Dual-functional TAT-P1 could efficiently deliver DIG-3 by transfection into mouse lung cells to inhibit viral replication and also directly inhibit viral replication by preventing endosomal acidification. We confirmed that DIG-3 delivered by TAT-P1 in mice further improved the survivals of avian A(H7N7) or human A(H1N1) virus-infected mice.

## Results

**Construction of influenza DIG plasmids.** Influenza defective interfering PB2 (DI-PB2), DI-PB1, and DI-PA genes with large internal deletion were generated from the backbone of A/WSN/1933(H1N1) virus using fusion PCR. Each DI-PB2, DI-PB1, and DI-PA consisting of internal deletions were inserted into the phw2000 plasmid (Fig. 1a). When these plasmids of DI-PB2, DI-PB1, and DI-PA were co-transfected or individually transfected into 293T and A549 cells, 7–8 log copies per well of each DIG RNA was detected by RT-qPCR (Fig. 1b, c and Supplementary Fig. 1).

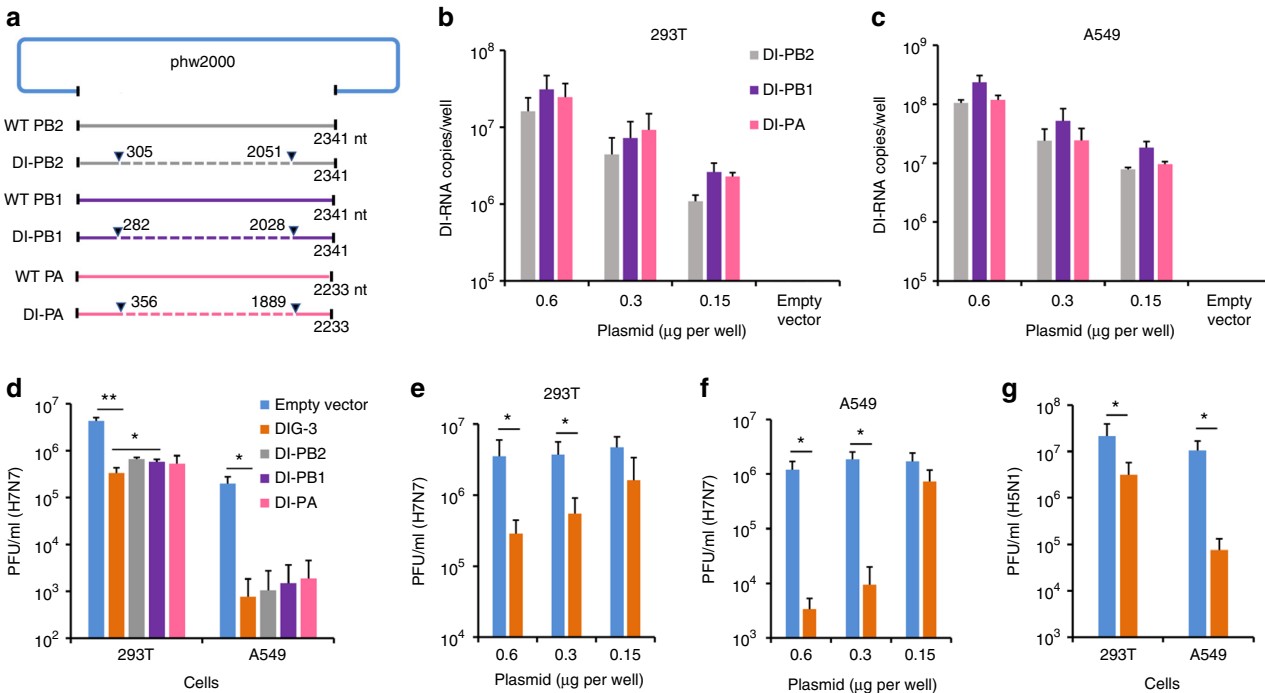

**Fig. 1** Construction and antiviral activity of defective interfering genes (DIG). **a** The plasmid construction of DI-PB2, DI-PB1, and DI-PA. The indicated sequences of shortened viral polymerase gene PB2, PB1, and PA were inserted into phw2000, respectively. Dotted lines indicate the internal deletion of wild-type (WT) viral polymerase genes. **b, c** DI RNA expression in 293T and A549 cells. The plasmids of DI-PB2, DI-PB1, and DI-PA were co-transfected into cells with the indicated concentrations. At 24 h post transfection, DI RNAs were extracted from cells and digested by DNase I for RT-qPCR. Empty vector was used as a negative control for RT-qPCR. **d** Anti-A(H7N7) virus activity of individual plasmid of DI-PB2, DI-PB1, and DI-PA or three combined plasmid DIG (DIG-3, 0.6 µg per well). **e, f** Dose-dependent anti-A(H7N7) virus activity of DIG-3 in 293T and A549 cells. **g** Anti-A(H5N1) virus activity of DIG-3. Empty vector phw2000 and plasmids with DIG were individually transfected to cells. At 24 h post transfection, cells were infected with A(H7N7) or A(H5N1) virus at MOI = 0.005 and cell supernatants were collected at 40 h post infection. Viral titers in the supernatants were detected by plaque assay. Data were presented as mean ± SD of three independent experiments. * Indicates $P < 0.05$. ** Indicates $P < 0.01$. $P$ values were calculated by the two-tailed Student's $t$ test

**DIG-transfected cells have lower viral replication**. Next, we evaluated the replication of influenza A virus in DIG transfected 293T and A549 cells. We chose A(H7N7) and A(H5N1) viruses for this in vitro assay because unlike human seasonal A(H1N1) pdm09 and A(H3N2) viruses, avian A(H7N7), and A(H5N1) viruses can replicate without trypsin in 293T or A549 cells. When 293T and A549 cells were transfected with plasmids of DI-PB2, DI-PB1, or DI-PA individually, the replication of H7N7 virus was reduced by more than 90% and 99%, respectively, compared with that of cells transfected with the empty vector (Fig. 1d). In 293T cells, the reduction of viral replication was significantly more pronounced when all three plasmids of DIG (DIG-3) were co-transfected together than that when only single DIG was transfected. Although, in A549 cells, there was no significant difference between DIG-3 and single DIG, we decided to perform subsequent experiments using co-transfected DIG-3. When increasing concentration of DIG-3 was used in transfection, the antiviral efficacy against A(H7N7) virus was improved in both 293T and A549 cells in a dose-dependent manner (Fig. 1e, f). DIG-3 also showed a significant anti-A(H5N1) virus activity in both 293T and A549 cells (Fig. 1g). Collectively, these results indicated that DIG-3 could significantly inhibit both A(H7N7) and A(H5N1) virus replication in different human cell lines.

**DIG-3 outcompetes full-length viral genes and generates DIV**. In order to identify whether DIG-3 could inhibit viral replication by generating DIV, we first confirmed that DIG-3 could significantly inhibit viral replication in A549-Dual KO-RIG-I cells,

which indicated that the antiviral activity of DIG-3 was not interferon dependent (Fig. 2a).

Next, we inoculated A(H7N7) virus in 293T cells that were pre-transfected with DIG-3 or the empty vector, and measured the viral RNA copies of full-length viral polymerase genes (PA, PB1, and PB2) and DI genes in cell supernatants. As shown in Fig. 2b–d, full-length PA, PB1, and PB2 RNA copies of A(H7N7) virus in the supernatants of DIG-3-transfected cells were more than 10-fold lower than those in the supernatants of empty vector-transfected cells, respectively. Importantly, RNA copies of DI-PA, DI-PB1, and DI-PB2 were 6–21-fold higher than those of full-length PA, PB1, and PB2 in the supernatants of DIG-3-transfected cells (Fig. 2b–d), indicating that significantly more DI RNAs than full-length viral RNAs were incorporated into virions to form DIV. To further confirm the generation of DIV in supernatants of DIG-3-transfected cells after wild-type A(H7N7) virus infection, we measured viral titers in the supernatants by plaque assay and HA assay and then compared the ratio of PFU and HA titer between DIG-3-transfected cells and empty vector-transfected cells (Fig. 2e). As DIV is not viable and cannot form plaques (Supplementary Fig. 2), the plaque assay measures only the wild-type plaque-forming virus without DIG, while HA assay can detect both of wild-type virus and virus containing DIG (DIV). The viral titer (PFU) in the supernatants of DIG-3-transfected 293T cells was <10% of that in the supernatants of empty vector-transfected cells. Since wild-type virus titers (PFU) correlated with HA titers in a linear fashion (Supplementary Fig. 3), it was expected that HA titer of virus in supernatants of DIG-3-transfected 293T cells was also <10% of viral titer in the

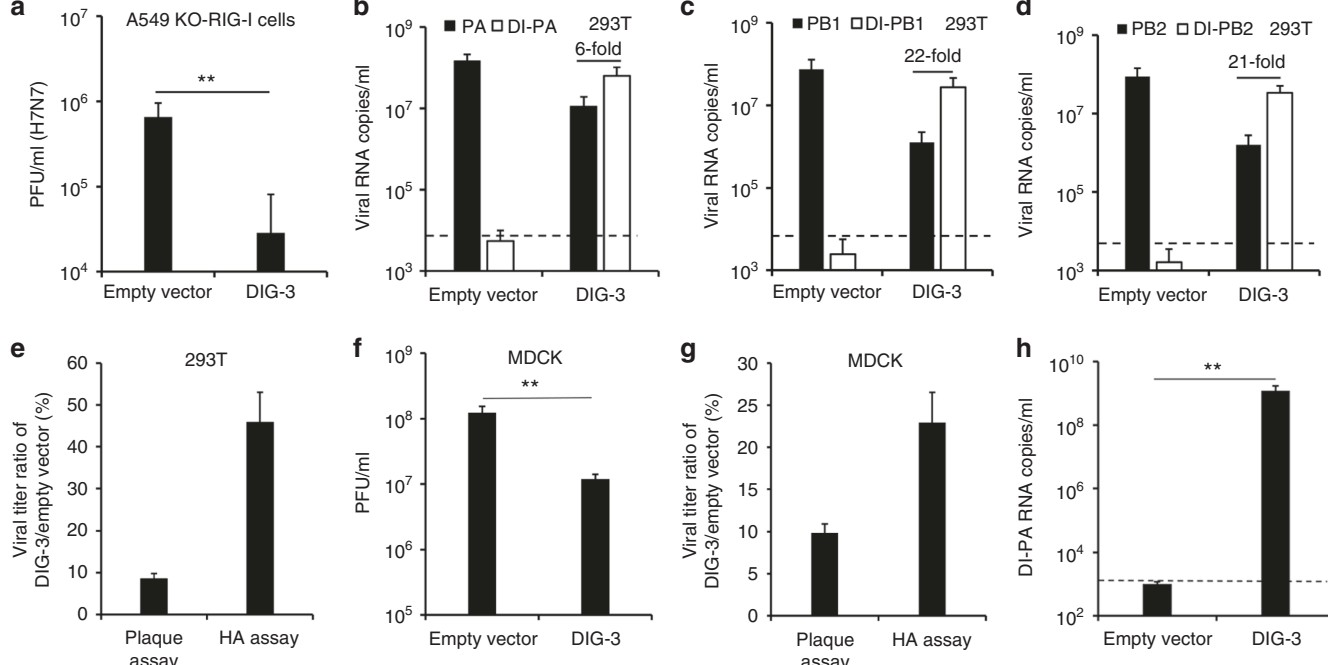

**Fig. 2** DIG could be packaged to generate defective interfering virus and competitively inhibit normal viral replication. **a** Antiviral activity of DIG-3 in A549-Dual KO-RIG-I cells. The A549-Dual KO-RIG-I cells were infected with A(H7N7) virus at 24 h post transfection of empty vector or DIG-3 (0.6 μg per well). Viral titers in cell supernatants of 40 h post infection were measured by plaque assay. **b–d** Full-length PA, full-length PB1, full-length PB2, DI-PA, DI-PB1, and DI-PB2 RNA levels in the supernatants of A(H7N7)-infected cells transfected with empty vector or DIG-3 before viral infection. **e** The percentage of DIG-3-treated virus compared with empty vector-treated virus in supernatants of 293T cells. Viral titers were measured by plaque assay and HA assay. **f** The passaged viral titers in supernatants of MDCK cells that were infected by the supernatant virus of A(H7N7)-infected 293T cells transfected with empty vector or DIG-3. **g** The virus titer ratio of DIG-3-treated virus compared with empty vector-treated virus in supernatants of MDCK cells. Viral titers were measured by plaque assay and HA assay. **h** The DI-PA RNA levels of passaged virus in the supernatants of MDCK cells. Dotted lines indicate the detection limit of RT-qPCR. Data were presented as mean ± SD of three independent experiments. ** Indicates $P < 0.01$. $P$ values were calculated by the two-tailed Student's $t$ test

supernatants of empty vector-transfected cells. However, HA titer of virus in the supernatants of DIG-3-transfected cells was about 45% of viral titer in the supernatants of empty vector-transfected cells, suggesting that virus with DIG was generated in supernatants. This is consistent with the result in Fig. 2b–d, which indicated that there were significantly higher DIG copies than full-length viral RNA copies in supernatants of DIG-3-transfected cells.

**DIV inhibits viral replication in non-transfected cells**. In Fig. 2b–e, we have shown that DIV was generated when wild-type influenza virus infected DIG-3-transfected cells. It would be important to know whether these newly generated DIV could subsequently inhibit the replication of wild-type virus in non-transfected cells. To this end, we collected the supernatant from A (H7N7)-infected 293T cells pre-transfected with DIG-3 and the supernatant from A(H7N7)-infected 293T cells pre-transfected with empty vector, and inoculated the supernatant viruses onto non-transfected MDCK cells at an MOI of 1. At 10 h post

infection, MDCK cells infected with the supernatant virus from DIG-3-transfected 293T cells had a significantly lower viral titer than that of MDCK cells infected with the supernatant virus from empty vector-transfected 293T cells (Fig. 2f). The generation of DIV in MDCK cell-passaged virus was further confirmed by the higher virus titer ratio (DIG-3/empty vector) in HA assay when compared with the virus titer ratio in plaque assay (Fig. 2g) and by detecting high DI-PA RNA copies in supernatants of MDCK cells (Fig. 2h). Therefore, our data demonstrated that DIG could be packaged to generate DIV when the DIG-3-transfected cells were infected with wild-type virus, and the resultant DIV could sustain the antiviral activity by competitively inhibiting wild-type viral replication in non-transfected cells.

**DIG-3 protects mice from lethal virus challenge**. To evaluate the in vivo antiviral efficacy of DIG-3, we tested prophylactic and therapeutic efficacy of DIG-3 and single DIG against influenza A virus infection in mice (Fig. 3a and Supplementary Fig. 4). The in vivo jetPEI, a commercially available polyethylenimine-based

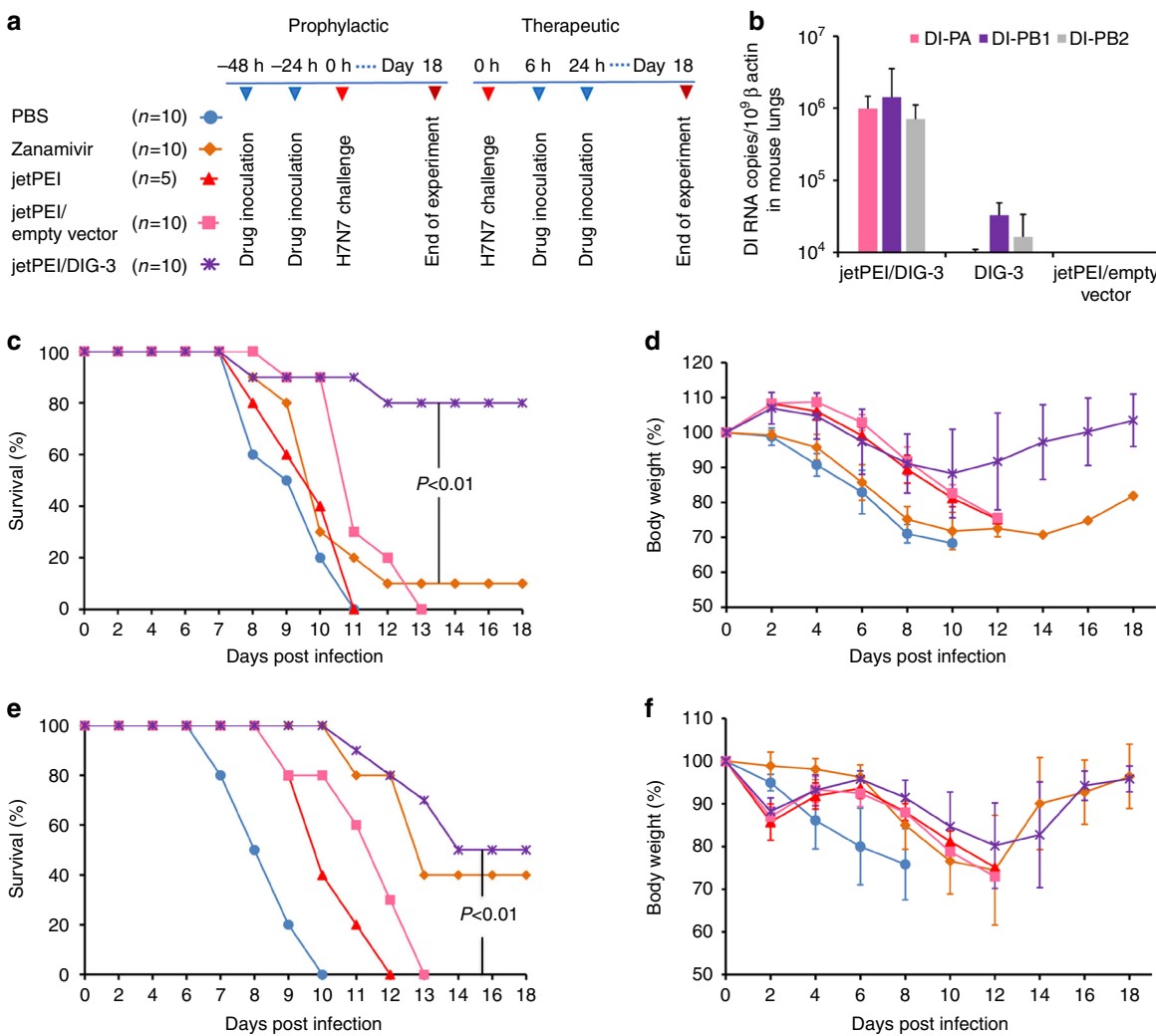

**Fig. 3** The jetPEI/DIG-3 could provide potent anti-A(H7N7) virus efficacy in mice. **a** Experimental design for evaluating antiviral efficacy of jetPEI/DIG-3 in mice. **b** Expression of DI-PA, DI-PB1, and DI-PB2 RNAs in mouse lungs when jetPEI/DIG-3 (5 μg per mouse), DIG-3, jetPEI/empty vector were intratracheally inoculated to mouse lungs. Three mice in each group were included. **c**, **d** Prophylactic efficacy of jetPEI/DIG-3 against A(H7N7) virus. **e**, **f** Therapeutic efficacy of jetPEI/DIG-3 against A(H7N7) virus. For prophylactic experiment, 40 μl of PBS (n = 10), zanamivir (50 μg in PBS, n = 10), jetPEI (0.7 μl in 5% glucose solution, n = 5), jetPEI/empty vector (0.7 μl/5.0 μg in 5% glucose solution, n = 10), and jetPEI/DIG-3 (0.7 μl/5.0 μg in 5% glucose solution, n = 10) were intratracheally inoculated to corresponding mice at 48 and 24 h before viral inoculation. For therapeutic experiment, PBS, zanamivir, jetPEI, jetPEI/empty vector and jetPEI/DIG-3 were intratracheally inoculated to corresponding mice at 6 and 24 h after viral inoculation. Survivals and body weight data were generated from 5 to 10 mice in each group with mean ± SD. P values were calculated by Gehan–Breslow–Wilcoxon test

vector, was used to deliver DIG-3 plasmids in vivo[23,24]. The jetPEI and DI-PA, DI-PB1, and DI-PB2 complex (jetPEI/DIG-3) were delivered intratracheally. DI-PA, DI-PB1, and DI-PB2 RNAs were successfully expressed in the lungs of transfected mice at 24 h post transfection (Fig. 3b).

We first evaluated the protective efficacy of DIG-3 and single DIG on infected mice (Supplementary Fig. 4). DIG-3 showed similar prophylactic and therapeutic efficacy as DI-PB2 (Supplementary Fig. 4a–d). The survivals (80%) of mice treated by DIG-3 or DI-PB2 were slightly higher than the survivals (70%) of mice treated by DI-PB1 and DI-PA as prophylaxis, but did not reach statistical significance (Supplementary Fig. 4a). The body weight loss of mice treated by DIG-3 or DI-PB2 were also about 5–10% less than that of mice treated by DI-PA and DI-PB1 from day 10 to day 14 (Supplementary Fig. 4b). For therapeutic treatment (Supplementary Fig. 4c–d), the survival rates (40–50%) and body weight change were comparable in infected mice treated by DIG-3 or single DIG. Next, we compared the prophylactic efficacy of DIG-3 and zanamivir in mice. The jetPEI/DIG-3, jetPEI/empty vector, jetPEI without DIG-3, zanamivir or PBS were administered intratracheally at 48 and 24 h before A(H7N7) infection. The survival rate of the jetPEI/DIG-3 group (80%) was significantly higher than that of all other control groups (≤10%) (Fig. 3c). The jetPEI/DIG-3 group also had significantly less body weight loss than those of zanamivir or PBS groups from day 6 to day 10 post infection (Fig. 3d). To compare the efficacy of DIG-3 and zanamivir as therapeutic treatment, jetPEI/DIG-3, jetPEI/empty vector, jetPEI, zanamivir or PBS were administered intratracheally at 6 and 24 h after A(H7N7) infection. Zanamivir and jetPEI/DIG-3 protected 40% and 50% of mice from lethal A(H7N7) virus challenge, respectively, with no statistically significant difference. The survival of the jetPEI/DIG-3 group was significantly higher than that of the jetPEI, jetPEI/empty vector, or PBS groups (Fig. 3e). The jetPEI/DIG-3 and zanamivir groups also had significantly less body weight loss than PBS group from day 6 to day 8 post infection (Fig. 3f). These results indicated that jetPEI/DIG-3 showed potent anti-A(H7N7) virus efficacy, which was better than zanamivir as prophylaxis and comparable to zanamivir as therapeutics against A(H7N7) virus infection in mice.

**Peptide TAT-P1 delivers plasmid DNA in vitro and in mice.** Although jetPEI/DIG-3 improved the survival rate of mice with lethal challenge by A(H7N7) virus, the survival rate was only 50% in terms of therapeutic treatment. In order to improve the effectiveness of DIG-3 as therapeutics, we investigated the use of a delivery peptide which also possesses antiviral activity. Previous studies showed that HIV-1 Tat peptide (TAT) conjugated with cationic peptides could enhance delivery of nucleic acids to cells[18,19,25,26]. In our previous study, a cationic peptide P9, derived from mouse β-defensin 4, has antiviral activity against A(H1N1)pdm09, A(H3N2), A(H5N1), and A(H7N7) viruses[22]. Thus, we designed three shorter derivatives of P9, namely P1, P2, and P3, and linked them to TAT (Supplementary Table 1). TAT-P1, TAT-P2, and TAT-P3 showed potent antiviral activity against A(H7N7) and A(H1N1)pdm09 virus, with IC$_{50}$ of <1.0 μg ml$^{-1}$ (Supplementary Table 2). TAT-P1 had the highest selective index (535) and was selected for subsequent experiments. When the antiviral activity of P1 and TAT was assessed separately (Supplementary Table 2), P1 retained the antiviral activity (IC$_{50}$ = 1.6 μg ml$^{-1}$), while TAT itself did not show any antiviral activity (IC$_{50}$ > 50.0 μg ml$^{-1}$). Our previous study showed that P9 bound to viral hemagglutinin and prevented endosomal acidification[22]. Here, we further confirmed that P1 and TAT-P1 could also bind to viral HA protein using ELISA and western blot assay (Fig. 4a and Supplementary Fig. 5).

Bafilomycin A1 (A1), P1 and TAT-P1 prevented endosomal acidification (Fig. 4b, c) and blocked viral RNP release into the nuclei (Fig. 4d, e), but not the P9-aci-1 (PA1)[22] which was a negative control peptide with similar sequence as P1 (Supplementary Table 1). However, P1 and TAT-P1 did not inhibit HA-mediated membrane fusion (Supplementary Fig. 6). Therefore, P1 and TAT-P1 exerted the antiviral activity through binding to HA and preventing endosomal acidification.

Next, the binding ability of TAT-P1 to plasmid DNA was evaluated with gel retardation assay (Supplementary Fig. 7a). Our data showed that TAT-P1 could bind and form complexes with DNA when the weight ratio (peptide:DNA) was >2. The sizes of peptide/DNA complexes were determined at various peptide/DNA weight ratios (Supplementary Fig. 7b). Particle sizes between 120 and 180 nm were formed when complexes were prepared in water with weight ratios from 2 to 8. The in vitro transfection efficiency of TAT-P1/pLuciferase (TAT-P1/pLuc) was evaluated in 293T cells. With the increase of weight ratio (TAT-P1:pLuc) from 2 to 8, the transfection efficiency increased (Fig. 4f). The transfection efficiency of TAT-P1/pLuc was significantly higher than that of TAT/pLuc, P1/pLuc, and mock-transfected cells.

We further determined whether TAT-P1 could efficiently deliver plasmid DNA into mouse lung cells. TAT-P1/pCMV-Luc or jetPEI/pCMV-Luc was administered intratracheally and luciferase expression was measured at 24 h post transfection. Luciferase expression in mouse lungs transfected with TAT-P1/pCMV-Luc was significantly higher than that in mouse lungs mock-transfected with TAT-P1/jetPEI without DNA, but was comparable to that of jetPEI/pCMV-Luc (Fig. 4g). When TAT-P1 and plasmid of DI-PA (TAT-P1/DI-PA) was intratracheally inoculated to mouse lungs, DI-PA RNA expression in mouse lungs was significantly higher than that in mouse lungs transfected with DI-PA without TAT-P1, but was comparable to that of jetPEI/DI-PA (Fig. 4h). Therefore, TAT-P1 is an effective system for in vivo transfection of plasmids. These results illustrated that TAT-P1 could directly exert antiviral activity by preventing endosomal acidification and also efficiently transfect plasmids in vivo.

**TAT-P1/DIG-3 shows anti-A(H7N7) virus activity in mice.** To evaluate the prophylactic efficacy of TAT-P1/DIG-3 against viral infection in mice, different doses of TAT-P1/DIG-3 were intratracheally administered to mice at 48 and 24 h before A(H7N7) virus infection. The survival rate of mice was increased in a dose-dependent manner and mice receiving DIG-3 at 5.0 μg per dose had 100% survival (Supplementary Fig. 8). Next, we evaluated the antiviral efficacy of mice receiving DIG-3 at 5.0 μg per dose and compared the result to those of zanamivir-treated and untreated controls. The survival of mice treated with TAT-P1/DIG-3 was significantly higher than that of mice treated with zanamivir, TAT-P1, or PBS (Fig. 5a). Body weight loss on days 6–10 post infection (Fig. 5b), viral titers (Fig. 5c), and the pro-inflammatory cytokine IL-6 (Fig. 5d) were significantly reduced in the TAT-P1/DIG-3-treated mice when compared with those of mice treated with zanamivir or PBS.

For therapeutic study, TAT-P1/DIG-3 and jetPEI/DIG-3 were intratracheally administered to mice at 6 and 24 h post infection. TAT-P1/DIG-3-treated mice achieved a survival of 90% (Fig. 5e), and was significantly higher than that of mice treated with zanamivir (40%, $P < 0.05$, Gehan–Breslow–Wilcoxon test) or jetPEI/DIG-3 (40%, $P < 0.05$, Gehan–Breslow–Wilcoxon test). TAT-P1 could confer 30% protection to infected mice. Body weight loss on days 6–8 (Fig. 5f), viral titers (Fig. 5g), and pro-inflammatory cytokine IL-6 expression (Fig. 5h) in lung tissues were significantly reduced in mice treated with TAT-P1/DIG-3

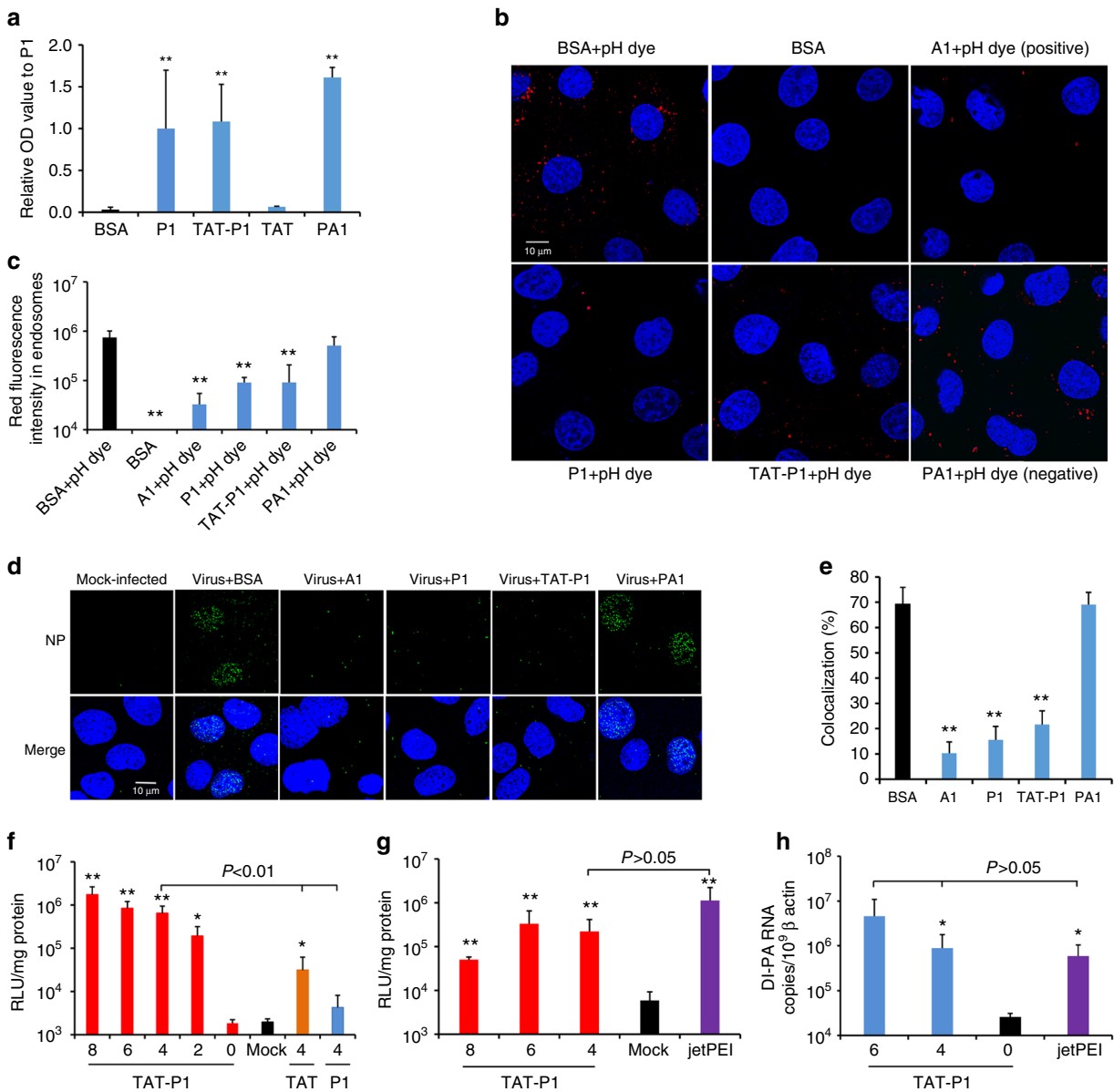

**Fig. 4** The antiviral mechanism and transfection efficiency of TAT-P1. **a** P1 and TAT-P1 could bind to H1N1 glycoprotein HA1 in an ELISA assay. OD values were normalized to P1 as 1. ** Indicates $P < 0.01$ when compared with BSA. **b** P1 and TAT-P1 could prevent endosomal acidification. Red dots (pHrodo™ dextran) indicate pH lower than 6.0 in endosomes. P9-aci-1 (PA1), which did not prevent endosomal acidification, was a negative peptide control for P1. Blue color indicates nuclei. Live cell images were taken by confocal microscope (original magnification ×400, scale bar = 10 μm). **c** The quantification of red fluorescence of endosomal acidification in MDCK cells when cells were treated by BSA, bafilomycin A1 (A1), P1, TAT-P1, or PA1 corresponding to **b**. Ten microscope fields of each sample were included for quantification. ** Indicates $P < 0.01$ when compared with BSA + pH dye group. **d** P1 and TAT-P1 could prevent viral RNP release into nuclei. MDCK cells were infected with 1 MOI of A(H1N1) virus treated with or without drugs. Images of NP (green) and nuclei (blue) were taken at 3.5 h post infection (scale bar = 10 μm). **e** Percentages of NP colocalized to nuclei. The SD was determined from multiple microscope fields including ~ 200 cells for each sample. ** Indicates $P < 0.01$ when compared with BSA. **f** Transfection efficiency of peptide/pLuc in 293T cells. Numbers indicate weight ratios of corresponding peptide:pLuc. Mock means cells treated with TAT-P1 without DNA. Data were presented as mean ± SD of three independent experiments. **g** Transfection efficiency of pCMV-Luc transfected by TAT-P1 or jetPEI in mouse lungs. Luciferase expression in mouse lungs was normalized to 1 mg protein. Mock means mice treated with TAT-P1 or jetPEI without DNA. **h** Transfection efficiency of plasmid of DI-PA transfected by TAT-P1 or jetPEI in mouse lungs. Data were presented as mean ± SD of ≥3 mice in each group. * Indicates $P < 0.05$. ** Indicates $P < 0.01$ when compared with Mock in **f** and **g** or when compared with "0" (naked DNA) in **h**. $P$ values were calculated by the two-tailed Student's $t$ test

when compared with mice treated with PBS. IL-6 was significantly lower in mice treated by TAT-P1/DIG-3 than that in mice treated by jetPEI/DIG-3.

**TAT-P1/DIG-3 protects mice from A(H1N1)pdm09 virus infection**. To evaluate the antiviral efficacy of TAT-P1/DIG-3 against seasonal influenza virus, the prophylactic and therapeutic

antiviral efficacy of TAT-P1/DIG-3 against A(H1N1)pdm09 virus were tested (Fig. 6). In prophylactic experiment (Fig. 6a–d), the survival of A(H1N1)pdm09-infected mice treated with TAT-P1/DIG-3 (50%) was lower than that of mice treated with zanamivir (90%), almost reaching statistical significance $(P = 0.06,$ Gehan–Breslow–Wilcoxon test) (Fig. 6a). TAT-P1/DIG-3 and zanamivir significantly reduced body weight loss on days 6–10

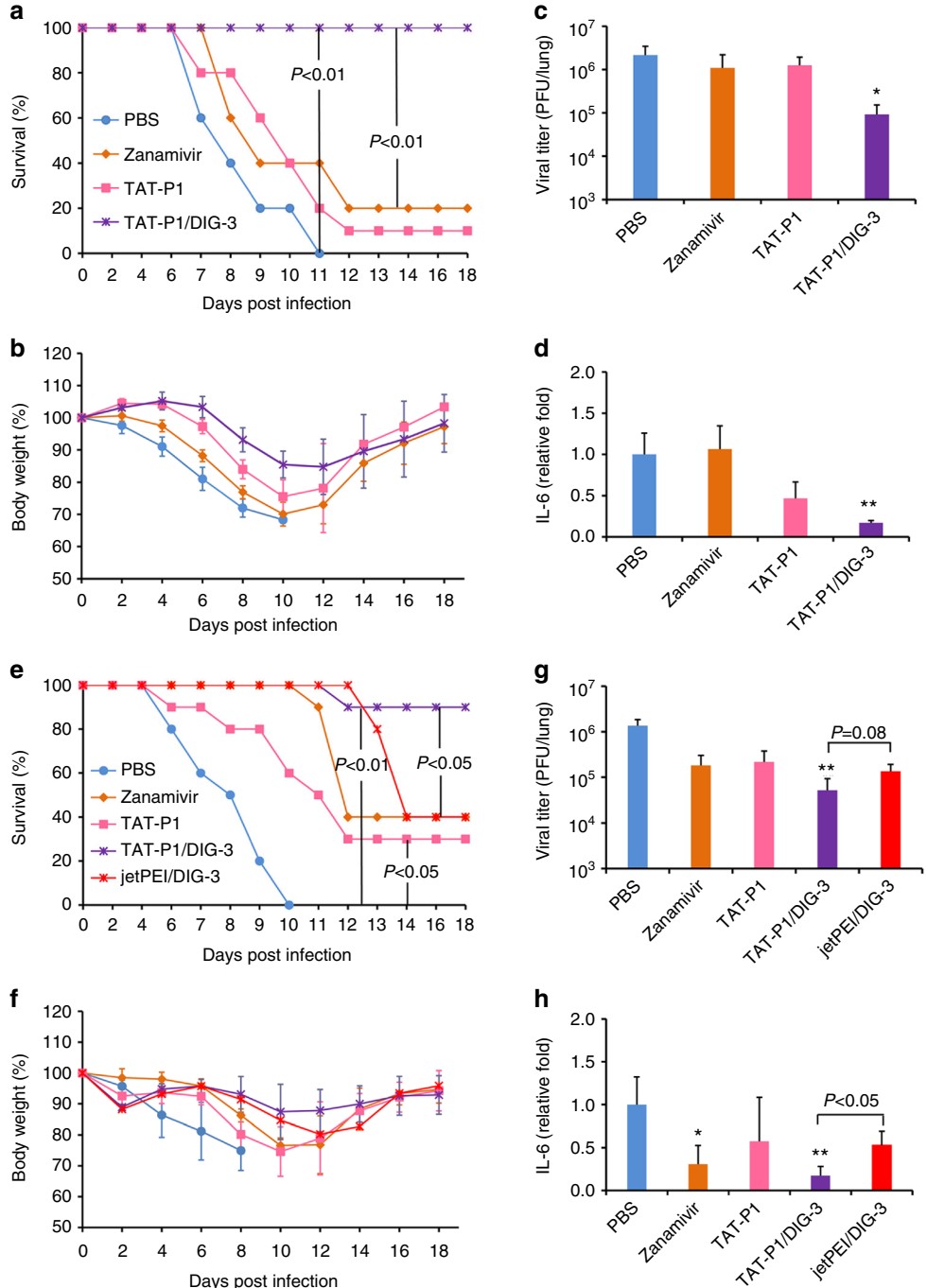

**Fig. 5** TAT-P1/DIG-3 could provide prophylactic and therapeutic protection against A(H7N7) virus infection in mice. **a–d** Prophylactic efficacy of TAT-P1/ DIG-3 against A(H7N7) virus. **e–h** Therapeutic efficacy of TAT-P1/DIG-3 against A(H7N7) virus. For prevention, 40 µl of PBS (n = 10), zanamivir (50 µg in PBS, n = 10), TAT-P1 (20 µg in distilled water, n = 10), and TAT-P1/DIG (20 µg/5 µg in distilled water, n = 10) were intratracheally inoculated to corresponding mice at 48 and 24 h before viral inoculation. For therapeutic experiment, PBS (n = 10), zanamivir (n = 10), TAT-P1 (n = 10), TAT-P1/DIG (n = 10), or jetPEI/DIG (0.7 µl/5.0 µg in 5% glucose solution, n = 5) were intratracheally inoculated to corresponding mice at 6 and 24 h after viral inoculation. Viral titers and IL-6 in mouse lungs were detected at day 4 post infection with mean ± SD of three mice in each group. The expression of IL-6 was normalized to PBS group. * Indicates $P < 0.05$. ** Indicates $P < 0.01$ when compared with PBS group. $P$ values were calculated by Gehan–Breslow–Wilcoxon test for survivals and by the two-tailed Student's $t$ test for viral titer and IL-6 analysis

(Fig. 6b), viral titers (Fig. 6c), and IL-6 expression in mice (Fig. 6d) when compared with PBS. In therapeutic experiment (Fig. 6e–h), the survival of mice treated with TAT-P1/DIG-3 (93%) or zanamivir (90 %) was significantly better than that of mice treated with PBS (0%) or jetPEI/DIG-3 (20%) (Fig. 6e). TAT-P1/DIG-3 significantly reduced body weight loss on days 4–8 (Fig. 6g), viral titers (Fig. 6f), and IL-6 expression (Fig. 6h) in

mouse lungs when compared with PBS group. Collectively, these data of TAT-P1/DIG-3 anti-A(H7N7) and anti-A(H1N1)pdm09 virus in mice demonstrated that the dual-functional TAT-P1 could directly inhibit viral infection in mice and also efficiently deliver DIG-3 into mouse lungs to exert sustained antiviral activity for prophylactic and therapeutic treatment. Even though the prophylactic protection of TAT-P1/DIG-3 on A(H1N1)

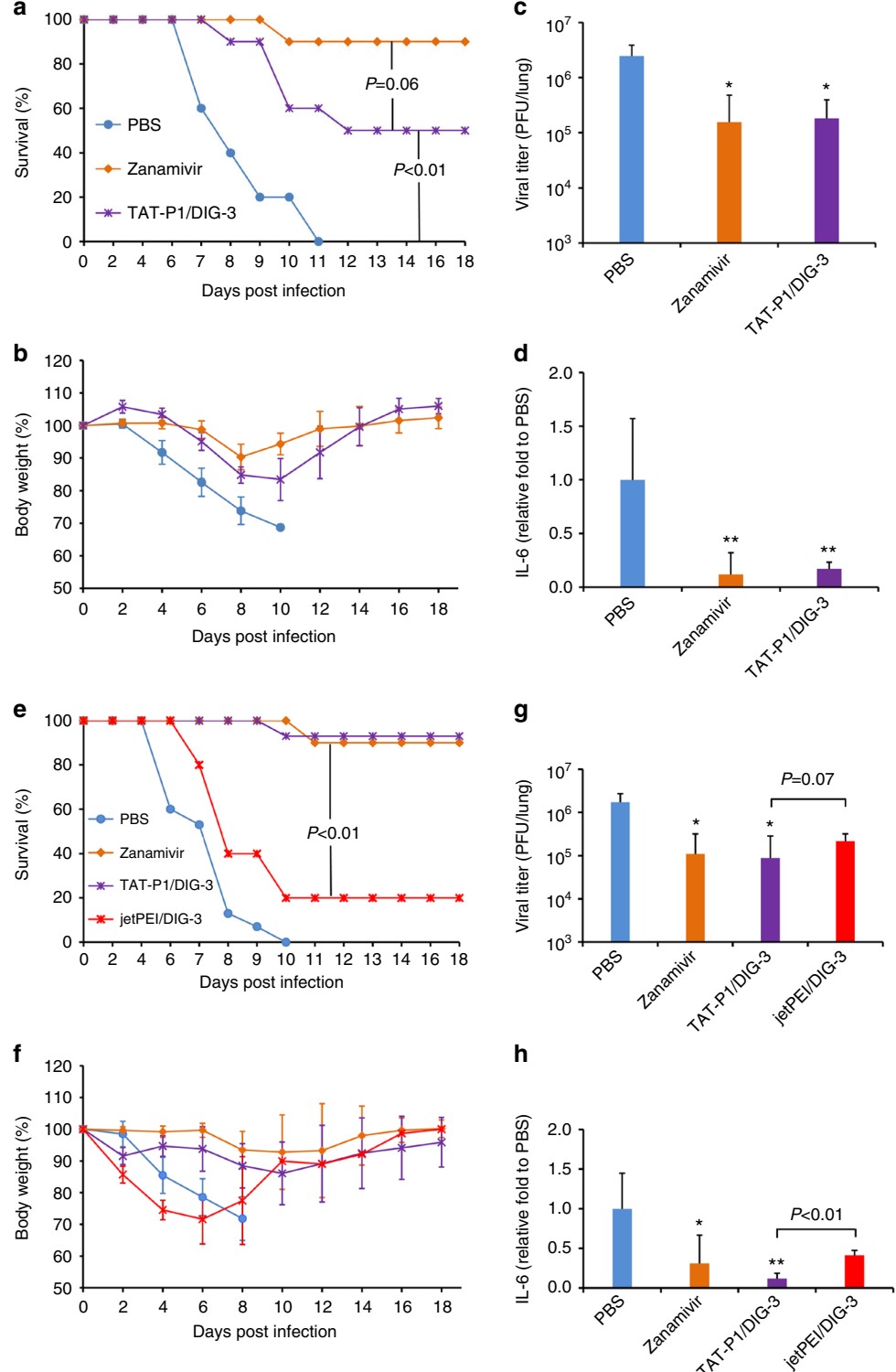

**Fig. 6** TAT-P1/DIG-3 could provide prophylactic and therapeutic protection against A(H1N1)pdm09 virus infection in mice. **a–d** Prophylactic efficacy of TAT-P1/DIG-3 against A(H1N1)pdm09 virus. **e–h** Therapeutic efficacy of TAT-P1/DIG-3 against A(H1N1)pdm09 virus. For prevention, 40 μl of PBS ($n = 10$), zanamivir (50 μg in PBS, $n = 10$), and TAT-P1/DIG-3 (20 μg/5 μg in distilled water, $n = 10$) were intratracheally inoculated to corresponding mice at 48 and 24 h before viral inoculation. For therapeutic experiment, PBS ($n = 15$), zanamivir (10), TAT-P1/DIG-3 ($n = 15$) or jetPEI/DIG-3 (20 μg/5 μg in 5% glucose solution, $n = 5$) were intratracheally inoculated to corresponding mice at 6 h and 24 h after viral inoculation. Viral titers and IL-6 in mouse lungs were detected at day 4 post infection with mean ± SD of >3 mice in each group. The expression of IL-6 was normalized to PBS group. * Indicates $P < 0.05$. ** Indicates $P < 0.01$ when compared with PBS group. $P$ values were calculated by Gehan–Breslow–Wilcoxon test for survivals and by the two-tailed Student's $t$ test for viral titer and IL-6 analysis

pdm09-infected mice was not as effective as zanamivir, the antiviral efficacy of TAT-P1/DIG-3 was comparable to that of zanamivir against A(H1N1)pdm09 virus in mice for therapeutic treatment and was significantly better than that of zanamivir against A(H7N7) virus for prophylactic and therapeutic treatment. The lower protection of zanamivir on A(H7N7)-infected mice than that of zanamivir on A(H1N1)pdm09-infected mice might be due to the reduced susceptibility of A(H7N7) virus to zanamivir (Supplementary Fig. 9).

**P1 improves the prophylactic efficacy of TAT-P1/DIG-3**. The survival rate of A(H1N1)pdm09-infected mice transfected with TAT-P1/DIG-3 before viral challenge was 50%. We hypothesized that the prophylactic antiviral efficacy of TAT-P1/DIG-3 against A(H1N1)pdm09 virus could be further improved by increasing the transfection efficiency of TAT-P1/DIG-3. Previous studies showed that inhibition of endosomal acidification by ATPase inhibitor can increase the transfection efficiency of TAT[27,28]. Using a luciferase assay, we demonstrated that the P1 peptide with inhibitory activity against endosomal acidification (Fig. 4c, d) could improve the transfection efficiency of TAT-P1/pLuc in 293T cells (Fig. 7a) and in mouse lungs (Fig. 7b). As a control, PA1 peptide, which cannot inhibit endosomal acidification (Fig. 4c, d), did not improve the transfection efficiency of TAT-P1/pLuc in 293T cells or in mouse lungs (Fig. 7a, b). The improvement of transfection efficiency by P1 was further confirmed by images of In Vivo Imaging System (Fig. 7c) and DI-PA RNA expression in mouse lungs (Fig. 7d).

We then evaluated whether the addition of P1 peptide could improve the survival of A(H1N1)pdm09-infected mice transfected with TAT-P1/DIG-3. P1 and TAT-P1/DIG-3 were intratracheally inoculated to mice at 48 and 24 h before A (H1N1)pdm09 virus infection. PA1 was used as a negative control. As shown in Fig. 7e, P1 combined with TAT-P1/DIG-3 (TAT-P1/DIG-3 + P1) conferred an improved survival of 70% when compared with TAT-P1/DIG-3 (40%). TAT-P1/DIG-3 + P1 also significantly inhibited viral replication in mouse lungs when compared with that of A(H1N1)pdm09-infected mice treated with TAT-P1/DIG-3 (Fig. 7f). PA1 peptide could not improve the survival (40%) of A(H1N1)pdm09-infected mice treated with TAT-P1/DIG-3 and did not significantly inhibit viral replication in mice when compared with that of A(H1N1)pdm09-infected mice treated with TAT-P1/DIG-3. In conclusion, these data indicated that additional P1 could further enhance the transfection efficiency of TAT-P1/DIG-3 in vivo, which conferred an improved protection on A(H1N1)pdm09-infected mice.

## Discussion

In this study, we sought to investigate the use of DIG in the treatment of avian and seasonal influenza virus infections. Firstly, we demonstrated that DIG-3 can significantly inhibit the replication of A(H7N7) and A(H5N1) viruses in 293T and A549 cells and protect mice from lethal A(H7N7) and A(H1N1)pdm09 virus challenge as prophylaxis or therapeutics. Secondly, we improved the treatment efficacy of DIG-3 by using a novel delivery vector

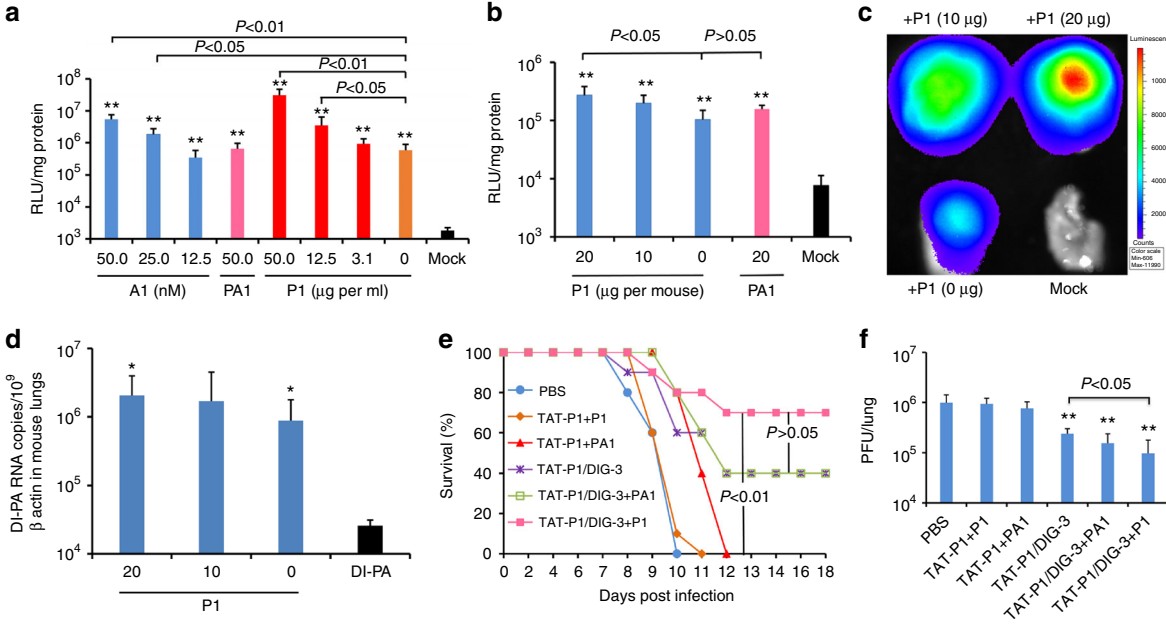

**Fig. 7** The transfection efficiency and antiviral activity of TAT-P1 with DIG-3 could be increased by P1 peptide. **a** Transfection efficiency of TAT-P1/pLuc increased by additional ATPase inhibitor (bafilomycin A1) or P1 peptide. Before TAT-P1/pLuc (2.0 μg/0.5 μg) complex was added to 293T cells for transfection, the indicated concentrations of bafilomycin A1 (A1, nM), P1 peptide (μg per ml), or PA1 peptide (μg per ml) were added to cell culture media. Mock indicates cells treated with A1 or TAT-P1 without DNA. Data were presented as mean ± SD of three independent experiments. **b** Transfection efficiency of TAT-P1/pCMV-Luc increased by P1 in mouse lungs. **c** Representative In Vivo Imaging System image showed increased luciferase expression by P1 in mouse lungs. TAT-P1/pCMV-Luc (20 μg/5 μg) with additional P1 (20 μg, 10 μg, or 0 μg) or PA1 (20 μg) were inoculated to mouse lungs at 48 and 24 h before measuring bioluminescence signal or taking bioluminescence image. Mock indicates mouse lungs inoculated with TAT-P1 + P1 without DNA. **d** The RNA expression of DI-PA increased by P1 in mouse lungs. TAT-P1/DI-PA (20 μg/5 μg) with additional P1 (20 μg, 10 μg, or 0 μg) were inoculated to mouse lungs at 48 and 24 h before detecting RNA expression. Data were presented as mean ± SD of ≥3 mice. * Indicates $P < 0.05$ and ** Indicates $P < 0.01$ when normalized to Mock (**a**, **b**) or naked plasmid of DI-PA (**d**). **e**, **f** The prophylactic efficacy of TAT-P1/DIG-3 + P1 against A(H1N1)pdm09 virus. PBS (n = 10), TAT-P1 + P1(n = 10) or TAT-P1 + PA1 (20 μg + 20 μg, n = 5), TAT-P1/DIG-3 (20 μg/5 μg, n = 10), TAT-P1/DIG-3 + PA1 (20 μg/5 μg + 20 μg, n = 5), or TAT-P1/DIG-3 + P1 (20 μg/5 μg + 20 μg, n = 10) were intratracheally inoculated to corresponding mice at 48 and 24 h before viral inoculation. ** Indicates $P < 0.01$ when compared with PBS. $P$ values were calculated by Gehan–Breslow–Wilcoxon test for survival analysis and by the two-tailed Student's t test for analysis in **a**, **b**, **d** and **f**

TAT-P1, which also has intrinsic antiviral effect via the inhibition of endosomal acidification. In our experiments, TAT-P1/DIG-3 conferred significantly better mouse survival than that of zana-mivir when used as prophylaxis or therapeutics against A(H7N7) virus in mice. Thirdly, the addition of P1 peptide to mouse lungs can further improve the transfection efficiency of TAT-P1/DIG-3 and the survival of mice. Thus, we have successfully developed a prophylactic and therapeutic strategy using TAT-P1/DIG-3, which interferes with the replication of a diverse subtypes of influenza virus at two steps within one life cycle of virus as observed in this scenario (Fig. 8). This novel dual-functional delivery system, TAT-P1, directly exerts antiviral activity and transfects DIG efficiently into cells to competitively inhibit wild-type viral replication. Development of resistance against DIG will be unlikely as DIG does not act on a particular viral target[16].

DIV consisting of one or more defective genes has been studied as a strategy for treating influenza virus infection[13,14]. DIV could provide prophylactic and therapeutic protection on infected mice and ferrets[13,15], which might be affected by host immunity but not reliant on the interferon response in mice[12,29]. However, there are concerns of DIV generating new reassortants and neutralizing antibody[17]. In contrast, DIG only consists of defec-tive genes, which will only express DI RNAs without any full-length viral RNA and will not generate new self-replicable reas-sortants. Furthermore, DIG will not induce the neutralizing antibody because no protein product is required for DIG-induced protection[12]. In this study, we illustrated that DI RNAs expressed by transfected plasmids in cells could significantly inhibit viral replication in an interferon independent manner and be packaged to generate DIV (Fig. 2a–e), which could outcompete wild-type

virus[9] and showed sustained antiviral activity in non-transfected MDCK cells (Fig. 2f–h).

One major obstacle of using DIG in vivo is whether DIG can be safely and efficiently delivered to the cells at the site of infection. Viral vectors, polyethylenimine derivatives, and peptides have been used for gene delivery in vivo. Although viral vectors have been used in clinical trials for treating primary immunodeficiency diseases[30], cancer[31], HIV[32] and influenza virus infections[33,34], there are concerns about genomic instability, immunogenicity, and toxicity in humans[31,35,36]. Polyethylenimine derivatives have been used in animals for DNA transfection[23,24,37], but the application of polyethylenimine-based vectors in humans may be limited by their intrinsic tendency to disrupt cell membranes and mitochondrial membrane[38,39]. On the other hand, peptides have been considered as promising delivery vectors in humans because of the low toxicity and the absence of toxic metabolites[40,41]. Short peptides have been used clinically in humans as antiviral, anti-bacterial and anti-cancer drugs for many years[40,42,43]. HIV-TAT peptide, which penetrates cells in a receptor-independent man-ner[44], is an effective delivery peptide vector of protein and DNA through caveolae/lipid-raft-mediated endocytosis[45], micro-pinocytosis[27], clathrin-mediated endocytosis[46], and endocytosis-independent pathways[47]. Increasing evidence has shown that TAT combining with other peptides could increase the trans-fection efficiency in vitro and in vivo[19,21,25]. Furthermore, the transfection efficiency of TAT could be enhanced through increasing the endosomal escape by ATPase inhibitor (chlor-oquine) which disrupts endosomes by preventing endosomal acidification[27]. However, the effective concentrations of chlor-oquine (~100 μM) for inhibiting endosomal acidification are

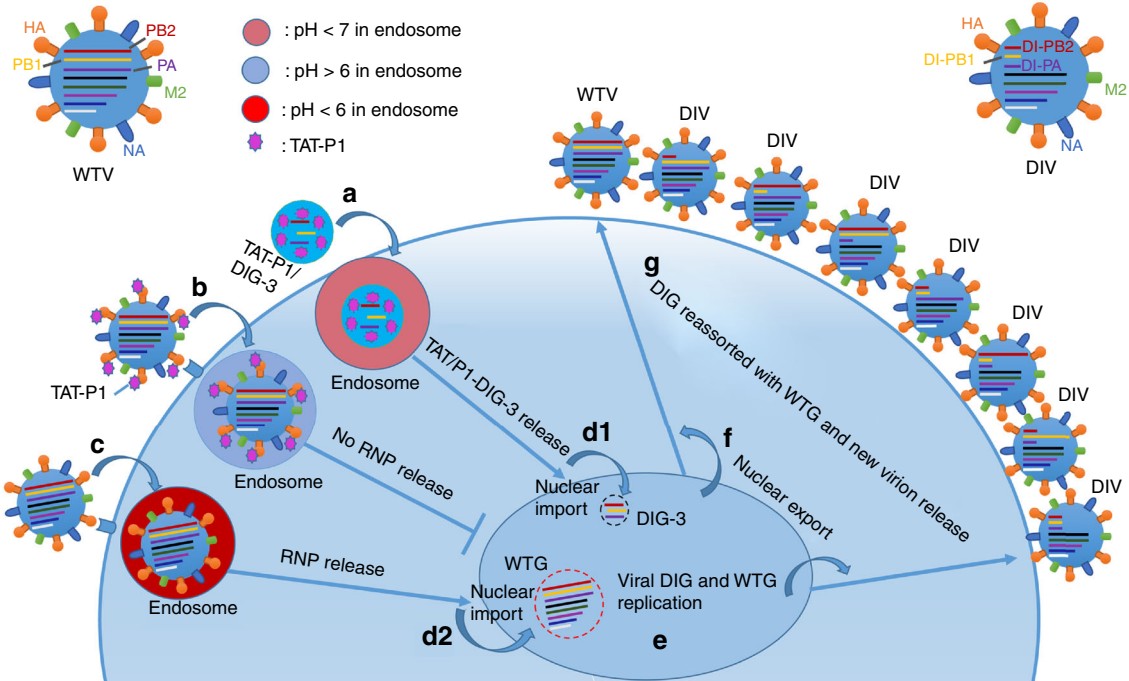

**Fig. 8** Schematic model of antiviral mechanism of TAT-P1/DIG-3 in one viral life cycle. When cells are transfected with TAT-P1/DIG-3 and infected with influenza virus, **a** TAT/P1-DIG-3 are internalized into cells by endocytosis; **b** Virus can be bound with TAT-P1 and internalized into cells by endocytosis. TAT-P1 can prevent endosomal acidification to block the release of viral RNP into the cytoplasm and then nuclei; **c** Some virus particles that are not bound by TAT-P1 might be internalized into cells by endocytosis and undergo endosomal acidification. **d** TAT-P1/DIG-3 released from endosomes are imported into nuclei (d1) and viral RNPs released from endosomes are imported into nuclei (d2). **e** Viral DIG and wild-type genes (WTG) replicate in nuclei. **f** Newly produced DIG and WTG are exported to cytoplasm. **g** Viral DIG and WTG are reassorted in cytoplasm and move to the cell membrane to be incorporated into new virions. Seven types of DIV and one type of wild-type virus (WTV) are released from cells to start new virus life cycle. The newly generated DIV can have sustained antiviral activity to competitively inhibit WTV replication in non-transfected cells

extremely toxic for cells in vitro and is almost near the lethal concentration in humans (~200 μM)[27,48].

Here, we aimed to develop a peptide vector TAT-P1, which is expected to have less safety concerns[36]. We chose the endosomal acidification inhibitor P1 peptide to combine with TAT for several reasons. Firstly, P1 possesses antiviral activity against both seasonal and avian influenza A viruses. Secondly, P1 can inhibit endosomal acidification with low cytotoxicity. P1 peptide could significantly enhance the transfection efficiency of plasmid DNA at a concentration (12.5 μg ml$^{-1}$) far below the CC$_{50}$ (>400 μg ml$^{-1}$) (Fig. 7a and Supplementary Table 2). The high transfection efficiency of TAT-P1/DIG-3 in combination with the direct antiviral activity of TAT-P1 allows the TAT-P1/DIG-3 treatment to exert the immediate antiviral activity by TAT-P1 and the sustained anti-influenza activity by DIG-3 in infected mice with low possibility to cause resistance[16].

Mice treated by TAT-P1/DIG-3 had poorer survival in the prophylactic setting (Figs. 6a and 7e) than that of therapeutic setting (Fig. 6e). One possibility is that TAT-P1 was not able to exert antiviral activity to protect mice when administered to mouse lungs at 24 h before viral inoculation in the prophylactic setting.

Challenged mice still had body weight loss despite treatment by DIG-3 encoded by jetPEI or TAT-P1. This is probably due to the relatively high lethal dose of influenza virus (4 LD$_{50}$) used for mouse challenge. In addition, DIG could competitively inhibit wild-type virus replication but could not completely abolish viral replication in mice. Thus, mice would be infected and lose body weight even when DIG-3 was transfected. However, since the DIG-3 could reduce viral replication, the body weight loss in the DIG-3-treated mice was much less severe than that of infected mice in the negative control groups.

In summary, we have demonstrated a dual-functional system with both gene delivery and antiviral ability in vivo. This dual-functional TAT-P1 with DIG complex has broad antiviral activity with a low likelihood of inducing antiviral resistance. We have established a concept for developing transfection vectors which may have wide applications in gene antiviral strategies including the delivery of antiviral gene/siRNA to combat influenza and non-influenza viruses for treating viral respiratory diseases. Further studies with TAT-peptide variants should be performed to determine the best TAT-peptide variants for DIG-3 delivery.

## Methods

**Cell culture and viruses**. Madin Darby canine kidney (MDCK, CCL-34), 293T (CRL-3216) and A549 (CCL-185) cells obtained from ATCC (Manassas, VA, USA) were cultured in Dulbecco minimal essential medium (DMEM) supplemented with 10% fetal bovine serum (FBS), 100 IU ml$^{-1}$ penicillin and 100 μg ml$^{-1}$ streptomycin. A549-Dual KO-RIG-I (InvivoGen, USA, Cat# A549d-korigi) cells were cultured in DMEM supplemented with 10% FBS, 100 IU ml$^{-1}$ penicillin, 100 μg ml$^{-1}$ streptomycin, 10 μg ml$^{-1}$ blasticidin, 100 μg ml$^{-1}$ zeocin, and 2 mM L-glutamine. The virus strains used in this study included A/Hong Kong/415742/2009[49], A/Hong Kong/415742Md/2009 (H1N1) (a highly virulent mouse-adapted strain)[50], A/Vietnam/1194/2004 (H5N1)[22], and A/Netherlands/219/2003 (H7N7)[51]. For in vitro experiments, viruses were cultured in MDCK cells. For animal experiments, viruses were cultured in eggs as described previously[52].

**Construction of plasmids**. Plasmids containing the full-length sequence of wild-type A/WSN/1933 PA, PB1, and PB2 genes[53] were used as the template to generate defective interfering PA, PB1, and PB2 genes with internal deletion by fusion PCR[54]. Short gene segments at 5′ end and 3′ end of each of the genes were amplified with gene-specific primers (Supplementary Table 3) designed by Primer Premier 5.0. For our DI genes, we selected the 5′ and 3′ ends of polymerase gene segments because these regions contain the packaging signals[55,56]. Furthermore, we chose 282–356 nt and 291–345 nt in 5′ and 3′ ends because previous studies showed that DI-PA, DI-PB1, and DI-PB2 genes from 291 to 617 nt could be isolated from infected mouse lungs[57], and the DIG with 317 nt in the 5′ end and with total length of 585 nt showed highest antiviral activity[56]. The amplified short gene fragments in the 5′ and 3′ ends were fused by fusion PCR to generate DI-PA, DI-PB1, and DI-PB2 genes using six pairs of primers (Supplementary Table 3). The

fused DI-PA, DI-PB1, and DI-PB2 genes (Supplementary Table 4) were inserted into BsmBI/BsaI sites of phw2000 vector to generate plasmids of DI-PA, DI-PB1, and DI-PB2, respectively. The DNA sequences of the constructed plasmids with DIG were verified by Sanger sequencing.

**Antiviral activity assay of DIG in cells**. For in vitro antiviral experiments, plasmids of DI-PA, DI-PB1, DI-PB2, and empty vector phw2000 were transfected into 293T and A549 cells by Lipofectamine 3000 reagent according to the manufacturer's instructions (Invitrogen, Cat# 1857483). After 24 h transfection, cells were washed with PBS and were inoculated with 1000 PFU of A(H7N7) or A(H5N1) virus in DMEM for infection and culture. Supernatant was collected at 40 h post infection. Viral titers were determined using plaque assay as we described previously[51].

**Viral RNA extraction and reverse transcription quantitative PCR**. Viral RNA was extracted by Viral RNA Mini Kit (QIAGEN, Cat# 52906, USA) according to the manufacturer's instructions. Extracted RNA were treated with DNase I (QIAGEN, Cat# 79254, USA) according to the manufacturer's protocol and purified by RNeasy Mini Kit (QIAGEN, Cat# 74106, USA) to exclude plasmid DNA contamination. Real-time RT-qPCR was performed as we described previously[22]. RNA was reverse transcribed to cDNA using primer Uni-12 and PrimeScript II 1st Strand cDNA synthesis Kit (Takara, Cat# 6210A) using GeneAmp® PCR system 9700 (Applied Biosystems, USA). The cDNA was then amplified using specific primers (Supplementary Table 5) for DI-PA, DI-PB1, DI-PB2 and wild-type H7N7 PA, PB1, PB2 using LightCycle® 480 SYBR Green I Master (Roach, USA). For quantitation, 10-fold serial dilutions of standard plasmid equivalent to 10$^1$ to 10$^6$ copies per reaction were prepared to generate the calibration curve. Real-time qPCR experiments were performed using LightCycler® 96 system (Roche, USA).

**Design and synthesis of peptides**. P1, TAT, and the fusion peptides TAT-P1, TAT-P2, and TAT-P3 were designed as shown in Supplementary Table 1 and synthesized by ChinaPeptide (Shanghai, China). The purity of all peptides was >95%. The purity and mass of each peptide were verified by HPLC and mass spectrometry.

**Cytotoxicity assay**. Cytotoxicity of peptides was determined by the detection of 50% cytotoxic concentration (CC$_{50}$) using a tetrazolium-based colorimetric MTT assay as we described previously[22]. Briefly, MDCK and 293T cells were seeded in 96-well cell culture plate at an initial density of 2 × 10$^4$ cells per well in DMEM supplemented with 10% FBS and incubated for overnight. Cell culture media were removed and then DMEM with various concentrations of peptides and 1% FBS were added to each well. After 24 h incubation at 37 °C, MTT solution (5 mg ml$^{-1}$, 10 μl per well) was added to each well. After incubation at 37 °C for 4 h, 100 μl of 10% SDS in 0.01 M HCl was added to each well. After further incubation at room temperature with shaking overnight, the plates were read at OD$_{570}$ using Victor$^{TM}$ X3 Multilabel Reader (PerkinElmer, USA). Cell culture wells without peptides were used as the experiment control and medium only served as a blank control.

**Plaque reduction assay for antiviral peptides**. Antiviral activity of peptides was measured using a plaque reduction assay as we described previously[22]. Peptides were dissolved in 30 mM phosphate buffer (PB) containing 24.6 mM Na$_2$HPO$_4$ and 5.6 mM KH$_2$PO$_4$ at a pH of 7.4[22]. Peptides or bovine serum albumin (BSA, 0.4–50.0 μg ml$^{-1}$) were premixed with A(H7N7) or A(H1N1)pdm09 viruses in phosphate buffer at room temperature. After 1 h of incubation, peptide-virus mixture was transferred to MDCK cells. At 1 h post infection, cells were washed with PBS once, and 1% low melting agar was added to cells. Cells were fixed using 4% formalin at 40 h post infection for A(H7N7) virus and 60 h post infection for A(H1N1)pdm09 virus. Crystal blue (0.1%) was added for staining, and the number of plaques was counted.

**ELISA assay**. Peptides (0.1 μg per well) dissolved in H$_2$O were coated onto ELISA plates and incubated at 4 °C overnight. Then, 2% BSA was used to block plates at 4 °C overnight. For HA binding, 2 μg ml$^{-1}$ in PB buffer of HA1 (Sino Biological Inc., Cat# 11055-V08H4) was incubated with peptides at 37 °C for 1 h. The binding abilities of peptides to HA1 protein were determined by incubation with rabbit anti-His-HRP (Invitrogen, Cat# R93125, 1: 2000) at room temperature for 30 min. The reaction was developed by adding 50 μl of TMB single solution (Life Technologies, Cat# 002023) for 15 min at 37 °C and stopped with 50 μl of 1 M H$_2$SO$_4$. Readings were obtained in an ELISA plate reader (Victor 1420 Multilabel Counter; PerkinElmer) at 450 nm.

**Western blot assay**. Peptide samples (1 μg) were loaded to SDS-PAGE and transferred to the polyvinylidene difluorid (PVDF) membrane. The transferred PVGF membrane was blocked by 10 % skimmed milk overnight and then incubated with HA1 (2 μg ml$^{-1}$) at room temperature for 1 h, followed by incubation with rabbit-IgG anti-HA (Sino Biological Inc. Cat# 11055-RP02, 1:4000) for 1 h to detect peptide-HA1 binding. Next, Goat anti-rabbit IgG-HRP (Invitrogen, Cat# 656120, 1:6000) was used as the secondary antibody to detect the binding at room

temperature for 1 h. Finally, immunoreactive bands were visualized by Luminata Classico Western HRP Substrate (Millipore, Cat# WBLUC0500).

**Endosomal acidification analysis in live cells**. Endosomal acidification was detected with a pH-sensitive dye (pHrodo Red dextran, Invitrogen, Cat#P10361) according to the manufacturer's instructions as previously described but with slight modification[22]. First, MDCK cells were treated with P1 (25.0 µg ml$^{-1}$), TAT-P1 (3.1 µg ml$^{-1}$), bafilomycin A1 (100.0 nM), P9-aci-1 (PA1, 25.0 µg ml$^{-1}$), or BSA (25.0 µg ml$^{-1}$) at 4 °C for 15 min. Second, MDCK cells were added with 100 µg ml$^{-1}$ of pH-sensitive dye and DAPI and then incubated at 4 °C for 15 min. Before taking images, cells were further incubated at 37 °C for 15 min and then cells were washed twice with PBS. Finally, PBS was added to cells and images were taken immediately with confocal microscope (Carl Zeiss LSM 700, Germany).

**Nucleoprotein (NP) immunofluorescence assay**. MDCK cells were seeded on cell culture slides and were infected with A(H1N1)pdm09 virus at 1 MOI pre-treated with BSA (25.0 µg ml$^{-1}$), bafilomycin A1 (50.0 nM), P1 (25.0 µg ml$^{-1}$), TAT-P1 (5.0 µg ml$^{-1}$), or PA1 (25.0 µg ml$^{-1}$). After 3.5 h post infection, cells were fixed with 4% formalin in PBS for 1 h, and permeabilized with 0.2 % Triton X-100 in PBS for 5 min. Cells were washed with PBS and then blocked with 5% BSA at room temperature for 1 h. Cells were incubated with mouse IgG anti-NP (Millipore, Cat# 2817019, 1:600) at room temperature for 1 h and then washed with PBS for next incubation with secondary antibody goat anti-mouse IgG Alexa-488 (Life Technologies, Cat# 1752514, 1:600) at room temperature for 1 h. Finally, cells were washed with PBS and stained with DAPI. Images were taken by confocal microscope (Carl Zeiss LSM 700, Germany).

**Polykaryon assay**. The 293T cells were seeded into 24-well plates at $2 \times 10^5$ cells per well. After overnight culture, the cells were transfected with phw2000-H7N7-HA plasmid (0.6 µg per well) using Lipofectamine 3000 (Invitrogen, Cat# 1857483) following the manufacturer's instructions. At 24 h after transfection, the transfection medium was replaced by DMEM containing BSA (50.0 µg ml$^{-1}$), P1 (50 µg ml$^{-1}$), TAT-P1 (10 µg ml$^{-1}$), or FA-617[58] (25 µM) and cells were incubated at 37 °C for 20 min. Polykaryon formation was induced by exposing cells to a low pH DMEM (pH 5.0) containing the corresponding concentrations of drugs at 37 °C for 10 min. The low pH DMEM medium was replaced with fresh DMEM containing 10% FBS and cells were incubated at 37 °C for 3 h. Finally, cells were fixed with 4% formalin in PBS and stained with Giemsa (Sigma). Images were taken by micro-scope at ×200 magnification.

**Particle size measurement**. According to the previous study[59], peptide/DNA complexes were prepared at various weight ratios. Peptide solution and plasmid DNA solution were prepared separately in distilled water. Equal volumes of peptide and plasmid DNA solution were mixed together to give a final volume of 4 µl containing 0.5 µg of plasmid DNA. After leaving the complexes for 15 min at room temperature and diluting the 4 µl complexes to 50 µl in distilled water, the particle diameter of the complexes was measured by DynaPro® Plate Reader (WYATT, USA).

**Gel retardation assay**. According to the previous study[59], peptide/DNA complexes were prepared at various ratios with 0.5 µg plasmid DNA in 4 µl distilled water. After leaving the complexes for 15 min at room temperature, the samples were loaded into a 1% w/v agarose gel containing ethidium bromide nucleic acid stain. Gel electrophoresis was run in TBE buffer at 100 V for 30 min and the gel was visualized under the ultraviolet (UV) illumination.

**In vitro luminescence analysis**. Peptide/DNA complexes were prepared at various weight ratios with 0.5 µg plasmid DNA in 4 µl distilled water. After incubating the complexes for 15 min at room temperature, the 293T cells in 24-well plate were transfected with the complexes including 0.1 µg of each pHW2000 plasmid encoding the PA, PB1, PB2, NP, and the mini-genome of pPoLI-fluc-RT (pLuc, the firefly luciferase reporter)[51]. At 24 h after transfection, luminescence was measured using Luciferase assay system (Promega, Cat# E1910) with a Victor X3 Multilabel reader (PerkinElmer, USA). The luminescence reading was normalized to 1 mg protein.

**In vivo bioluminescence analysis**. Peptide with pCMV-Cypridina Luc (pCMV-Luc, ThermoFisher, Cat# RF233236) complexes were prepared at various weight ratios with 15 µg plasmid DNA in 60 µl distilled water. After leaving the complexes for 15 min at room temperature, two doses of complexes were intratracheally inoculated to mouse lungs at 48 and 24 h before measuring the luciferase expression in lung tissues. The jetPEI/pCMV-Luc (2.1 µl/15.0 µg) complexes were prepared according to the manufactory protocol as a positive control (Polyplus Transfection, Cat# 201–10G). Mice inoculated with peptide or jetPEI only were used as the negative control. For detecting bioluminescence signal, mouse lung tissues were homogenized and centrifuged at $14,500 \times g$ for 5 min. The supernatant was used to analyze the luciferase protein expression by Cypridina luciferase flash assay kit (ThermoFisher, Cat# 16168). The luciferase expression level in mouse

lungs was normalized to 1 mg protein. For in vivo bioluminescence imaging, mouse lungs were taken out and then substrate was added to lungs for taking image by IVIS® Spectrum In Vivo Imaging System (PerkinElmer, USA).

**In vivo DI RNA expression analysis**. TAT-P1/DIG complexes were prepared with 5.0 µg plasmid DNA in 40 µl distilled water. After leaving the complexes for 15 min at room temperature, two doses of complexes were intratracheally inoculated to mouse lungs at 48 and 24 h before measuring the DIG RNA expression in lung tissues. The jetPEI/DIG (0.7 µl/5.0 µg) complexes were prepared according to the manufacturer's protocol as a positive control. Naked DIG was inoculated to mouse lungs as base line control. Mouse lung was harvested, flash-frozen, and stored in liquid nitrogen. Lung tissue was homogenized under liquid nitrogen and kept frozen at all times. Once tissue was completely homogenized in powder form, 1 ml TRIzol® Reagent (ThermoFisher, Cat# 15596026) was added to solubilize the tissue by gently mixing. Total RNA was firstly extracted by TRIzol® according to the manufacturer's instructions (Invitrogen, Cat# 87703). Next, the total RNA was further purified by RNeasy Mini Kit (Qiagen, Cat# 74106). In order to exclude the plasmid DNA contamination, all RNA samples were treated by DNase I (QIAGEN, Cat# 79254) according to the manufacturer's instructions and purified by RNeasy Mini Kit (Qiagen, Cat# 74106).

**Antiviral analysis of DIG-3 in mice**. BALB/c female mice (Laboratory Animal Unit, The University of Hong Kong), aged 12–16 weeks, were kept in biosafety level 3 laboratory and given access to standard pellet feed and water ad libitum. All experimental protocols followed the standard operating procedures of the approved biosafety level 3 animal facilities and were approved by the Committee on the Use of Live Animals in Teaching and Research of the University of Hong Kong[52]. The mouse adapted A(H1N1)pdm09 and A(H7N7) viruses were used for lethal chal-lenge in mice.

To evaluate the prophylactic efficacy, mice were intratracheally inoculated with 40 µl of PBS, zanamivir (50.0 µg in PBS), jetPEI (0.7 µl in 5% glucose solution), jetPEI/plasmids (0.7 µl/5.0 µg in 5% glucose solution), TAT-P1 (20.0 µg in distilled water), TAT-P1/plasmids (20.0 µg/5 µg in distilled water) at 48 h and 24 h before viral challenge. Next, mice were intranasally inoculated with 4 LD$_{50}$ of virus. For evaluation of the therapeutic efficacy, mice were intranasally inoculated with 4 LD$_{50}$ of virus. At 6 and 24 h post infection, mice were intratracheally inoculated with 40 µl of PBS, zanamivir (50.0 µg in PBS), jetPEI (0.7 µl in 5% glucose solution), jetPEI/plasmids (0.7 µl/5.0 µg in 5% glucose solution), TAT-P1 (20.0 µg in distilled water), or TAT-P1/plasmids (20.0 µg/5.0 µg in distilled water). Experimental mice were randomly allocated to each group. Survivals and general conditions were monitored by two investigators for 18 days or until death. Data were collected without exclusion. For viral titer and cytokine analysis, more than three mice in each group were sacrificed at day 4 after viral challenge.

**Statistical analysis**. The statistical significances of mouse survivals were analyzed by Gehan–Breslow–Wilcoxon test using GraphPad Prism 6 (San Diego, USA). The statistical significances of other experiments were calculated by the two-tailed Student's $t$ test. A $P$ value of <0.05 was considered to be statistically significant.

**Data availability**. All data that support the conclusions of the study are available from the corresponding author upon request.

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

## Acknowledgements

We thank Professor Bo-Jian Zheng (The University of Hong Kong) for his support and suggestions on this work. This work was partly supported by the donations of Michael Seak-Kan Tong, the Shaw Foundation Hong Kong, Richard Yu, and Carol Yu, Respiratory Viral Research Foundation Limited, Hui Ming, Hui Hoy and Chow Sin Lan Charity Fund Limited, and Chan Yin Chuen Memorial Charitable Foundation; and funding from the Health and Medical Research Fund of the Food and Health Bureau, Hong Kong Special Administrative Region Government; and the Collaborative Innovation Center for Diagnosis and Treatment of Infectious Diseases of the Ministry of Education of China.

## Author contributions

H.Z., K.K.W.T., H.C., and K.Y. designed this study and wrote the manuscript. H.Z., Q.D. X.Z., C.L., H.S., and S.Y. performed the experiments and analyzed the data. H.Z., K.K.W. T., H.C., J.Z., K.K., S.J., and K.Y. interpreted the results and revised the manuscript.

## Additional information

**Competing interests:** The authors declare no competing interests.

