## [Peer Review File · Nature Communications]

Editorial Note: An image has been redacted from this peer review file as indicated to protect copyright.

Reviewers' comments:

Reviewer #1 (Remarks to the Author):

This manuscript describes an analysis of three synthetically constructed defective interfering genomes (DIGs) derived from influenza A/WSN/1933(H1N1) virus with a focus on their ability to protect against disease after infection with influenza A(H7N7) or A(H1N1)pdm09 viruses. The data show very clearly that administration of the DIGs by transfection before or after infection provided significant protection assessed by survival curves. The use of transfection to deliver DIGs by transfection rather than packaged within a virus particle is novel and the authors use their expertise in this area to demonstrate enhancement of transfection efficiency in vivo. The data indicate that this approach, in principle, may have some clinical application subject to the serious issues of toxicity of the materials used. The preliminary characterization showing amplification of the DIGs in the presence of infectious virus, their packaging into virus particles and ability to inhibit virus replication in cell culture are all very clearly presented and provide a solid base for the remainder of the work.

My major concern with the data is that while the in vivo data are clear and demonstrate the protective capacity of the DIG3 combination. However, it is well known in the field that the ability of a DIG to interfere in vitro, as demonstrated here, does not correlate with the ability to protect from disease in vivo. By using only the DIG3 combination the authors do not know whether one or all of the DIGs have protective ability. This is an important question and the absence of data on the individual DIGs represents a major omission that should be addressed for the manuscript to be published. The data would be highly informative in efforts to understand what makes a protective DIG and would also be essential for this approach to be acceptable clinically as there remains doubt about what the active molecule actually is. This is sufficiently important that without the data the manuscript lacks an essential element required for acceptance.

Specific comments

The authors do not make clear the basis for the design of the DIGs that they used. For example were these based on known DIGs isolated from virus preparations or were they generated based on other data? It would be particularly interesting to indicate whether other DIGs had been made but were not protective as has been shown in the literature. The summary of construction in Figure 1a cannot be correct as the indicated lengths of the three virus genome segments (2280 nt for each of segments 1 and 2 and 2274 nt for segment 3) do not correspond to the actual segment lengths of the parental virus. This must be clarified as it is confusing.

While transfection of the DIGs independently and together inhibited virus replication in

cultured cells (Figure 1d), the statement in the paper that 'The reduction of viral replication was significantly more pronounced when all three plasmids of DIG (DIG-3) were co-transfected together into 293T cells' does not address the data showing that there was significantly greater inhibition overall in the A549 cells than in the 293 T cells (a ~1 log drop in 293T cells compared to a near 2 log drop in A549 cells). It is not clear why this would be the case and the authors may wish to comment briefly on possible reasons for the very significant difference that is observed. Is this due to differences in transfection efficiency and was this measured? Whatever the reason for differences the original statement that the inhibition is greater in 293T cells is not correct and should be corrected and their subsequent focus on 293T cell should be more clearly justified. In the further characterisation of the DIGs the authors looked at the levels of full length PA genome segment and the PA DIG RNA demonstrating that the DIG outcompeted the full length segment as would be expected. However given that subsequent work used all three DIGs for full characterisation similar data for the segment 1 and segment 2 DIGs should also be included to determine whether any one of the DIGs is more effective in competition than the others.

The authors rely significantly on survival data for the infected animals, supported by weight measurements. The weight measurements indicate that the surviving animals suffered some degree of deficit following infection and it would be valuable for the authors to comment on this.

When discussing Figure 6 the authors point out that the statistical analysis comparing the data from DIG3 and zanamivir treated mice had a p value of 0.06 which is described as not reaching statistical significance. They then later state that the two treatments were therefore comparable, despite what is clearly a large difference with a better performance with zanamivir treatment. The authors are overstating the case here and this must be altered. The convention that a p value of 0.05 represents significance is, of course, a common convention but it is simply that and failure to reach that level does not imply identity between data sets. The comments suggest a lack of full understanding of the use of statistical p values and the authors should moderate their comments significantly.

Figur3 8 is not referred to anywhere in the text. It does not seem necessary

Reviewer #2 (Remarks to the Author):

Zhao and co-workers present data on the superior efficiency of a novel dual-functional peptide with defective interfering genes in protecting mice against avian and seasonal influenza virus. Overall, the work is sound, experiments are carefully conducted, results clearly presented and discussed.

Still, a major point puzzles me and deserves further attention before publication. The authors claim the superior efficiency of Tat-linked interfering genes, and this is supported by data. However, in doing this, they refer to the available literature which, at least in part

(see references n. 18 and 25 of the present work), refer to Tat chimeric peptides (i.e. Tat-peptide variants with superior efficiency as compared to wild-type Tat peptide). Thus, I wonder whether it is correct to limit the experiments to Tat peptide, while claiming literature on Tat peptide variants. The authors should include those variants into their datasets and show whether they obtain (as it should be desirable) even better results.

Minor: Saolomone et al. Plos One 2013 should be cited along with ref. 25

Point-to-point responses

Reviewers' comments:

Reviewer #1 (Remarks to the Author):

This manuscript describes an analysis of three synthetically constructed defective interfering genomes (DIGs) derived from influenza A/WSN/1933(H1N1) virus with a focus on their ability to protect against disease after infection with influenza A(H7N7) or A(H1N1)pdm09 viruses. The data show very clearly that administration of the DIGs by transfection before or after infection provided significant protection assessed by survival curves. The use of transfection to deliver DIGs by transfection rather than packaged within a virus particle is novel and the authors use their expertise in this area to demonstrate enhancement of transfection efficiency in vivo. The data indicate that this approach, in principle, may have some clinical application subject to the serious issues of toxicity of the materials used. The preliminary characterization showing amplification of the DIGs in the presence of infectious virus, their packaging into virus particles and ability to inhibit virus replication in cell culture are all very clearly presented and provide a solid base for the remainder of the work.

My major concern with the data is that while the in vivo data are clear and demonstrate the protective capacity of the DIG3 combination. However, it is well known in the field that the ability of a DIG to interfere in vitro, as demonstrated here, does not correlate with the ability to protect from disease in vivo. By using only the DIG3 combination the authors do not know whether one or all of the DIGs have protective ability. This is an important question and the absence of data on the individual DIGs represents a major omission that should be addressed for the manuscript to be published. The data would be highly informative in efforts to understand what makes a protective DIG and would also be essential for this approach to be acceptable clinically as there remains doubt about what the active molecule actually is. This is sufficiently important that without the data the manuscript lacks an essential element required for acceptance.

Response:

We would like to thank the reviewer for providing the insightful comments. We agree with the reviewer that the ability of a DIG-3 or individual DIG to interfere viral replication *in vitro* may not correlate with the ability of DIG-3 or individual DIG to protect mice from infection *in vivo*. We have now followed the advice from the reviewer and performed mouse prophylaxis and treatment experiments using individual DIG and compared with DIG-3. We have found that there was no statistically significant difference in the survival of mice given individual DIG when compared to those which were given DIG-3 (Supplementary Fig. 4 in **line 164 of the revised manuscript**). Therefore, although DIG-3 had better antiviral activity than single DIG *in vitro*, this does not confer additional benefit *in vivo*. The lack of significant benefit of DIG-3 over individual DIG *in vivo* may be because DIG-3 has only slightly better antiviral effect (< 0.5 log reduction in viral load, Fig. 1d) than individual DIG. Although individual DIG has similar *in vivo* efficacy as DIG-3, we prefer to use DIG-3 for further development of this strategy for clinical use, because even if there is lack of antiviral effect of one DIG due to novel genetic reassortants, the antiviral activity of DIG-3 is unlikely to be affected.

Specific comments

The authors do not make clear the basis for the design of the DIGs that they used. For example were these based on known DIGs isolated from virus preparations or were they generated based on other data? It would be particularly interesting to indicate whether other DIGs had been made but were not protective as has been shown in the literature.

Response:

Thanks for the useful comment. Previously, Duhaut et al have shown that DIG (including DI-PA, DI-PB1 and DI-PB2) with 291-615 nt length could be isolated from infected mouse lungs¹. Duhaut *et al* have also demonstrated in a subsequent study that DIG with at least more than 150 nt in the 5' end is needed for DI virus with effective interfering activity, and DIG (POLI-317) with 317 nt in the 5' end and with total length of 585 nt showed the highest antiviral activity (see point-to-point response Fig. 1 below)². Thus, we designed DIGs (see point-to-point response Fig. 2 below) which are composed of around 282-356 nt in the 5' and 3' ends.

We have now stated these points clearly **in the page 17 from line 404 to line 409**, in the revised manuscript. Please also see our rationale described above on why we prefer to design DIG-3, rather than the individual DIG alone, as the potential clinically usable antiviral agent.

1. Duhaut, S.D. & Dimmock, N.J. Heterologous protection of mice from a lethal human H1N1 influenza A virus infection by H3N8 equine defective interfering virus: comparison of defective RNA sequences isolated from the DI inoculum and mouse lung. *Virology* **248**, 241-253 (1998).
2. Duhaut, S.D. & Dimmock, N.J. Defective segment 1 RNAs that interfere with production of infectious influenza A virus require at least 150 nucleotides of 5' sequence: evidence from a plasmid-driven system. *J Gen Virol* **83**, 403-411 (2002).

[redacted]

Point-to-point response Fig. 1. This is copied from the Fig. 2 of the published reference 2 which demonstrated the interference of virus replication of infectious WSN in the presence of a panel of

plasmids encoding defective segment 1 EQV RNAs with increasing lengths of 5' end sequence. Vero cell monolayers were transfected with 17 infectious WSN plasmids (each 0.5 µg) and 0 or 2 µg defective plasmid. The solid lines refer to the defective plasmids pPOLI-30, pPOLI- 80, pPOLI-150 and pPOLI-220, which each comprise 445 nt. Infectivity was determined in clarified medium sampled at 48 h post-transfection. In addition, interference by two other defective RNAs (**POLI-317** and POLI-d136) is shown for comparison. RNA expressed by pPOLI-317 (H7N7) is 585 nt in length and has 317 5' nt; points are indicated by plus. RNA from pPOLI-d136 (H3N8) is 436 nt in length and has 211 5' nt.

Point-to-point response Fig. 2 (which is our Fig. 1a in our revised manuscript) is the design of our DI-PB2, DI-PB1 and DI-PA. The indicated sequences of shortened viral polymerase gene PB2, PB1 and PA were generated by fusion PCR. Dotted lines indicate the internal deletion of wild-type (WT) viral polymerase genes. The DIG (POLI-317) published by Duhaut et al² was included here for reference.

The summary of construction in Figure 1a cannot be correct as the indicated lengths of the three virus genome segments (2280 nt for each of segments 1 and 2 and 2274 nt for segment 3) do not correspond to the actual segment lengths of the parental virus. This must be clarified as it is confusing.

Response:

We would like to thank the reviewer for pointing out this apparent difference in the genome length. In the original manuscript, we only included the coding sequence of PA, PB1 and PB2. We have revised the Fig. 1a which now includes both the coding sequences and the non-coding sequences (Fig. 1a of revised manuscript).

While transfection of the DIGs independently and together inhibited virus replication in cultured cells (Figure 1d), the statement in the paper that ‘The reduction of viral replication was significantly more pronounced when all three plasmids of DIG (DIG-3) were co-transfected together into 293T cells’ does not address the data showing that there was significantly greater inhibition overall in the A549 cells than in the 293 T cells (a ~1 log drop in 293T cells compared to a near 2 log drop in A549 cells). It is not clear why this would be the case and the authors may wish to comment briefly on possible reasons for the very significant difference that is observed. Is this due to differences in transfection efficiency and was this measured? Whatever the reason for differences the original statement that the inhibition is greater in 293T cells is not correct and should be corrected and their subsequent focus on 293T cell should be more clearly justified.

Response:

DIG showed more potent antiviral activity in A549 than that in 293T cells. There are two possible reasons. First, a higher level of DIG expression was detected in A549 cells when compared with that in 293T cells (Fig. 1b-c, Supplementary Fig. 1). Second, the virus replication was poorer in A549 cells than that of 293T cells (the virus titer was 10-fold lower in A549 cells than that of 293T cells, Fig.1d).

In the original manuscript (page 5, line 100), when we wrote the sentence “The reduction of viral replication was significantly more pronounced when all three plasmids of DIG (DIG-3) were co-transfected together into 293T cells”, we meant that the antiviral effect was more pronounced when compared with single DIG. We have now rewritten this sentence to “In 293T cells, the reduction of viral replication was significantly more pronounced when all three plasmids of DIG (DIG-3) were co-transfected together than that when only single DIG was transfected. Although, in A549 cells, there was no significant difference between DIG-3 and single DIG, ……” **(in page 5 from line 97 to line 100).**

In the further characterisation of the DIGs the authors looked at the levels of full length PA genome segment and the PA DIG RNA demonstrating that the DIG outcompeted the full length segment as would be expected. However given that subsequent work used all three DIGs for full characterisation similar data for the segment 1 and segment 2 DIGs should also be included to determine whether any one of the DIGs is more effective in competition than the others.

Response:

We have now tested the segment 1, segment 2 and segment 3 DIGs. DI-PB2 and DI-PB1 are more effective in out-competing DI-PA *in vitro*. The new data are now shown in Fig. 2b-d of the revised manuscript.

The authors rely significantly on survival data for the infected animals, supported by weight measurements. The weight measurements indicate that the surviving animals suffered some degree of deficit following infection and it would be valuable for the authors to comment on this.

Response:

Thank you for the comment. In our mouse infection model, we used a relatively high lethal dose

of influenza virus (4 LD₅₀) to challenge mice. In addition, DIG could competitively inhibit wild-type virus replication but could not completely abolish viral replication in mice. Thus, mice would be infected and lose body weight even when DIG-3 was transfected. However, since the DIG-3 could reduce viral replication, the body weight loss in the DIG-3-treated mice was much less severe than that of infected mice in the negative control groups. We have now added this point **in the Discussion section of page 15-16 from line 372 to line 378.**

When discussing Figure 6 the authors point out that the statistical analysis comparing the data from DIG3 and zanamivir treated mice had a p value of 0.06 which is described as not reaching statistical significance. They then later state that the two treatments were therefore comparable, despite what is clearly a large difference with a better performance with zanamivir treatment. The authors are overstating the case here and this must be altered. The convention that a p value of 0.05 represents significance is, of course, a common convention but it is simply that and failure to reach that level does not imply identity between data sets. The comments suggest a lack of full understanding of the use of statistical p values and the authors should moderate their comments significantly.

Response:

We agree with the reviewer. We have now revised the text to “Even though the prophylactic protection of TAT-P1/DIG-3 on A(H1N1)pdm09-infected mice was not as effective as zanamivir, the antiviral efficacy of TAT-P1/DIG-3 was comparable to that of zanamivir against A(H1N1)pdm09 virus in mice for therapeutic treatment and was significantly better than that of zanamivir against A(H7N7) virus for prophylactic and therapeutic treatment” in the **page 11 from line 271 to line 275.**

Figure 8 is not referred to anywhere in the text. It does not seem necessary

Response:

Figure 8 was used in the first paragraph of discussion section (in page 13 and line 309 of the original manuscript). We used this Fig. 8 to discuss the antiviral mechanism of TAT-P1/DIG3 (**in line 320 of page 13**). We hope that the reviewer and the editor can allow us to retain this figure because we believe that this figure can help the reader to understand the antiviral mechanism of TAT-P1/DIG-3 more easily.

Reviewer #2 (Remarks to the Author):

Zhao and co-workers present data on the superior efficiency of a novel dual-functional peptide with defective interfering genes in protecting mice against avian and seasonal influenza virus. Overall, the work is sound, experiments are carefully conducted, results clearly presented and discussed.

Still, a major point puzzles me and deserves further attention before publication. The authors claim the superior efficiency of Tat-linked interfering genes, and this is supported by data.

However, in doing this, they refer to the available literature which, at least in part (see references n. 18 and 25 of the present work), refer to Tat chimeric peptides (i.e. Tat-peptide variants with superior efficiency as compared to wild-type Tat peptide). Thus, I wonder whether it is correct to limit the experiments to Tat peptide, while claiming literature on Tat peptide variants. The authors should include those variants into their datasets and show whether they obtain (as it should be desirable) even better results.

Response:

We would like to thank the reviewer for the insightful comments. We agree with the reviewer that DIG delivered by other Tat-peptide variants may have better antiviral activity than that of DIG delivered by Tat-P1. However, since there are more than 20 Tat-peptide variants that have been published (see table below), it would take a long time for us to determine which Tat-peptide variant is better. We believe that by publishing our presently available data as a proof of concept at this stage would allow other groups to start testing the concept of DIG delivery by Tat-peptide and therefore the discovery of better Tat-peptide variants for this purpose. We have therefore amended **the last paragraph of the discussion** by adding “In addition, further studies with TAT-peptide variants should be performed to determine the best TAT-peptide variants for DIG-3 delivery.”

NO	Sequence	Author and publishing year	Data available	Purpose	Ref.
1	TAT-H6-K(C18)-YIGSR	Meng Z, et al, 2018	In vivo zebrafish	Delivery	3
2	TAT-NLS-REVD	Yang J, et al, 2016	In vitro	Delivery	4
3	Branched TAT	Jeong C, et al, 2016	In vitro	Delivery	5
4	TAT-Rp3	Favaro MT, et al, 2014	In vitro	Delivery	6
5	(Fmoc) ₂ KH ₇ -TAT	Han K, et al, 2013	In tumor	Anti-tumor delivery	7
6	NLS-TAT	Guo X, et al, 2013	In vitro	Delivery	8
7	TAT-PKKKRKV	Qu W, et al, 2013	In vivo Rat	Delivery	9
8	R9-TAT ₂	Lakshmanan M, et al, 2012	In vitro	Delivery	10
9	CM18-TAT	Salomone F, et al. 2012	In vitro	Delivery	11
10	Dimerized TAT	Kawabata A, et al, 2012	In vivo mouse	Anti-cancer delivery	12
11	TAT-Mu-AF	Govindarajan S, et al, 2012	In vivo mouse	Anti-tumor delivery	13
12	TAT-PKKKRKV	Yi WJ, et al, 2011	In vitro	Delivery	14
13	TAT-LK15	Saleh AF, et al, 2010	In tumor	Anti-tumor delivery	15
14	TAT-HMGB1A	Han JS, et al, 2009	In vitro	Delivery	16
15	TAT-10H	Lo SL, et al, 2008	In vivo rat	Delivery	17
16	rTAT-HMGB1A	Kim K, et al, 2008	In vitro	Delivery	18
17	YIGSR-TAT	Saleh AFA, et al, 2007	In vitro	Delivery	19
18	TAT-Mu	Rajagopalan R, et al, 2007	In vitro	Delivery	20
19	TAT-RGD	Renigunta A, et al, 2006	In vitro	Delivery	21
20	PolyTAT	Manickam, D.S, et al, 2005	In vitro	Delivery	22
21	Oligomers TAT	Rudolph, C, et al, 2003	In vitro	Delivery	23

Minor: Saolomone et al. *Plos One* 2013 should be cited along with ref. 25.

Response:

Thanks for suggestion. We cited this as reference 26 in line 194 of page 8.

Reference:

1. Duhaut, S.D. & Dimmock, N.J. Heterologous protection of mice from a lethal human H1N1 influenza A virus infection by H3N8 equine defective interfering virus: comparison of defective RNA sequences isolated from the DI inoculum and mouse lung. *Virology* **248**, 241-253 (1998).
2. Duhaut, S.D. & Dimmock, N.J. Defective segment 1 RNAs that interfere with production of infectious influenza A virus require at least 150 nucleotides of 5' sequence: evidence from a plasmid-driven system. *J Gen Virol* **83**, 403-411 (2002).
3. Meng, Z. et al. Enhanced gene transfection efficiency by use of peptide vectors containing laminin receptor-targeting sequence YIGSR. *Nanoscale* **10**, 1215-1227 (2018).
4. Yang, J. et al. Multitargeting Gene Delivery Systems for Enhancing the Transfection of Endothelial Cells. *Macromol Rapid Comm* **37**, 1926-1931 (2016).
5. Jeong, C., Yoo, J., Lee, D. & Kim, Y.C. A branched TAT cell-penetrating peptide as a novel delivery carrier for the efficient gene transfection. *Biomaterials research* **20**, 28 (2016).
6. Favaro, M.T. et al. Development of a non-viral gene delivery vector based on the dynein light chain Rp3 and the TAT peptide. *Journal of biotechnology* **173**, 10-18

(2014).

7. Han, K. et al. Synergistic gene and drug tumor therapy using a chimeric peptide. *Biomaterials* **34**, 4680-4689 (2013).
8. Guo, X., Chu, X., Li, W., Pan, Q. & You, H. Chondrogenic effect of precartilaginous stem cells following NLS-TAT cell penetrating peptide-assisted transfection of eukaryotic hTGFbeta3. *Journal of cellular biochemistry* **114**, 2588-2594 (2013).
9. Qu, W. et al. Peptide-based vector of VEGF plasmid for efficient gene delivery in vitro and vessel formation in vivo. *Bioconjugate chemistry* **24**, 960-967 (2013).
10. Lakshmanan, M., Kodama, Y., Yoshizumi, T., Sudesh, K. & Numata, K. Rapid and Efficient Gene Delivery into Plant Cells Using Designed Peptide Carriers. *Biomacromolecules* **14**, 10-16 (2013).
11. Salomone, F. et al. A novel chimeric cell-penetrating peptide with membrane-disruptive properties for efficient endosomal escape. *J Control Release* **163**, 293-303 (2012).
12. Kawabata, A. et al. Intratracheal administration of a nanoparticle-based therapy with the angiotensin II type 2 receptor gene attenuates lung cancer growth. *Cancer Res* **72**, 2057-2067 (2012).
13. Govindarajan, S. et al. Targeting human epidermal growth factor receptor 2 by a cell-penetrating peptide-affibody bioconjugate. *Biomaterials* **33**, 2570-2582 (2012).
14. Yi, W.J. et al. Enhanced nuclear import and transfection efficiency of TAT peptide-based gene delivery systems modified by additional nuclear localization signals. *Bioconjugate chemistry* **23**, 125-134 (2012).

15. Saleh, A.F. et al. Improved Tat-mediated plasmid DNA transfer by fusion to LK15 peptide. *J Control Release* **143**, 233-242 (2010).
16. Han, J.S., Kim, K. & Lee, M. A high mobility group B-1 box A peptide combined with an artery wall binding peptide targets delivery of nucleic acids to smooth muscle cells. *Journal of cellular biochemistry* **107**, 163-170 (2009).
17. Lo, S.L. & Wang, S. An endosomolytic Tat peptide produced by incorporation of histidine and cysteine residues as a nonviral vector for DNA transfection. *Biomaterials* **29**, 2408-2414 (2008).
18. Kim, K., Han, J.S., Kim, H.A. & Lee, M. Expression, purification and characterization of TAT-high mobility group box-1A peptide as a carrier of nucleic acids. *Biotechnol Lett* **30**, 1331-1337 (2008).
19. Saleh, A.F.A., Aojula, H.S. & Pluen, A. Enhancement of gene transfer using YIGSR analog of Tat-derived peptide. *Biopolymers* **89**, 62-71 (2008).
20. Rajagopalan, R., Xavier, J., Rangaraj, N., Rao, N.M. & Gopal, V. Recombinant fusion proteins TAT-Mu, Mu and Mu-Mu mediate efficient non-viral gene delivery. *The journal of gene medicine* **9**, 275-286 (2007).
21. Renigunta, A. et al. DNA transfer into human lung cells is improved with Tat-RGD peptide by Caveoli-mediated endocytosis. *Bioconjugate chemistry* **17**, 327-334 (2006).
22. Manickam, D.S., Bisht, H.S., Wan, L., Mao, G.Z. & Oupicky, D. Influence of TAT-peptide polymerization on properties and transfection activity of TAT/DNA polyplexes. *Journal of Controlled Release* **102**, 293-306 (2005).

23. Rudolph, C. et al. Oligomers of the arginine-rich motif of the HIV-1 TAT protein are capable of transferring plasmid DNA into cells. *Journal of Biological Chemistry* **278**, 11411-11418 (2003).

REVIEWERS' COMMENTS:

Reviewer #1 (Remarks to the Author):

The authors have fully addressed the comments and made appropriate changes to the manuscript. The additional information has consolidated the data presented originally.

Point-by-point response

REVIEWERS' COMMENTS:

Reviewer #1 (Remarks to the Author):

The authors have fully addressed the comments and made appropriate changes to the manuscript. The additional information has consolidated the data presented originally.

Response:

Thanks for this comment.